# Task Adaptation from Skills: Information Geometry, Disentanglement, and New Objectives for Unsupervised Reinforcement Learning

**Yucheng Yang**[1]**, Tianyi Zhou**[2]**, Qiang He**[3]**, Lei Han**[4]**, Mykola Pechenizkiy**[1]**, Meng Fang**[5,1]

[1]Eindhoven University of Technology, [2]University of Maryland, College Park,
[3]Ruhr University Bochum, [4]Tencent Robotics X, [5]University of Liverpool
{y.yang, m.pechenizkiy}@tue.nl, tianyi@umd.edu, Meng.Fang@liverpool.ac.uk

## Abstract

Unsupervised reinforcement learning (URL) aims to learn general skills for unseen downstream tasks. Mutual Information Skill Learning (MISL) addresses URL by maximizing the mutual information between states and skills but lacks sufficient theoretical analysis, e.g., how well its learned skills can initialize a downstream task's policy. Our new theoretical analysis in this paper shows that the diversity and separability of learned skills are fundamentally critical to downstream task adaptation but MISL does not necessarily guarantee these properties. To complement MISL, we propose a novel disentanglement metric LSEPIN. Moreover, we build an information-geometric connection between LSEPIN and downstream task adaptation cost. For better geometric properties, we investigate a new strategy that replaces the KL divergence in information geometry with Wasserstein distance. We extend the geometric analysis to it, which leads to a novel skill-learning objective WSEP. It is theoretically justified to be helpful to downstream task adaptation and it is capable of discovering more initial policies for downstream tasks than MISL. We finally propose another Wasserstein distance-based algorithm PWSEP that can theoretically discover all optimal initial policies.

## 1 Introduction

Reinforcement learning (RL) has drawn growing attention by its success in autonomous control (Kiumarsi et al., 2017), Go (Silver et al., 2016) and video games (Mnih et al., 2013; Vinyals et al., 2019). However, a primary limitation of the current RL is its high sample complexity. Inspired by the successful pretrain-finetune paradigm in other deep learning fields like natural language processing (Radford et al., 2019; Devlin et al., 2019) and computer vision (Henaff, 2020; He et al., 2020), there has been growing work studying the pretraining of RL. RL agent receives no task-related reward during pretraining and learns by its intrinsic motivations (Oudeyer & Kaplan, 2009). Some of these intrinsic motivations can help the agent to learn representations of the observations (Schwarzer et al., 2021) and some learn the dynamics model (Ha & Schmidhuber, 2018; Sekar et al., 2020). In this work, we focus on Unsupervised RL (URL) that learns a set of skills without external reward and the learned skills are expected to be quickly adapted to unseen downstream tasks.

A common approach for skill discovery of URL is Mutual Information Skill Learning (**MISL**) (Eysenbach et al., 2022) that maximizes the mutual information between state and skill latent (Eysenbach et al., 2019; Florensa et al., 2017; Hansen et al., 2020; Liu & Abbeel, 2021b). The intuition is that by maximizing this mutual information the choice of skills can effectively affect where the states are distributed so that these skills could be potentially used for downstream tasks. There are more algorithms using objectives modified on this mutual information. For example, Lee et al. (2019); Liu & Abbeel (2021b) added additional terms for better exploration, and Sharma et al. (2020); Park et al. (2022a) focus on modified input structure to prepare the agent for specific kinds of downstream tasks.

Despite the popularity of MISL, there has been little theoretical analysis of how well the MISL-learned skills can be applied as downstream task initializations. Previous work Eysenbach et al. (2022) has tried to analyze MISL but they consider an impractical downstream task adaptation procedure

that uses the average state distribution of all learned skills as initialization instead of directly using the learned skills. Therefore, it is still unclear how well the MISL-learned skills can be applied as downstream task initializations.

In this work, we theoretically analyze the connection between the properties of learned skills and their downstream task performance. Our results show that the diversity and separability of learned skills are fundamentally critical to downstream task adaptation. Separability, or the distinctiveness of skill distributions, is key for diverse skills. Without it, even a large number of skills may cover only a limited range resulting in limited diversity. The importance of diversity is empirically demonstrated in previous works (Eysenbach et al., 2019; Kim et al., 2021; He et al., 2022; Laskin et al., 2022). Our results also show that MISL alone does not necessarily guarantee these properties. To complement MISL, we propose a novel disentanglement metric that is able to measure the diversity and separability of learned skills. Our theoretical analysis relates the disentanglement metric to downstream task adaptation.

In particular, we introduce a novel disentanglemen metric "**L**east **SEP**arability and **IN**formativeness (**LSEPIN**)", which is directly related to the task adaptation cost from learned skills and complementary to the widely adopted mutual information objective of MISL. LSEPIN captures both the informativeness, diversity, and separability of the learned skills, which are critical to downstream tasks and can be used to design better URL objectives. We relate LSEPIN to **W**orst-case **A**daptation **C**ost (**WAC**), which measures the largest possible distance between a downstream task's optimal feasible state distribution and its closest learned skill's state distribution. Our results show increasing LSEPIN could potentially result in lower WAC.

In addition, we show that optimizing MISL and LSEPIN are essentially maximizing distances measured by KL divergences between state distributions. However, a well-known issue is that KL divergence is not a true metric, i.e., it is not symmetric and does not satisfy the triangle inequality. This motivates us to investigate whether an alternative choice of distance can overcome the limitations of MISL. Wasserstein distance is a symmetric metric satisfying the triangle inequality and has been feasibly applied for deep learning implementations (Arjovsky et al., 2017; Dadashi et al., 2020), so we investigate a new strategy that replaces the KL divergence in MISL with Wasserstein distance and exploits its better geometric properties for theoretical analysis. This leads to new skill learning objectives for URL and our results show that the objective built upon Wasserstein distance, "**W**asserstein **SEP**aratibility (**WSEP**)", is able to discover more potentially optimal skills than MISL. Furthermore, we propose and analyze an unsupervised skill-learning algorithm "**P**rojected **SEP**" (**PWSEP**) that has the favored theoretical property to discover all potentially optimal skills and is able to solve the open question of "vertex discovery" from Eysenbach et al. (2022).

Analysis of LSEPIN is complement to prior work to extend the theoretical analysis of MISL to practical downstream task adaptation, while the analysis of WSEP and PWSEP opens up a new unsupervised skill learning approach. Our results also answer the fundamental question of URL about what properties of the learned skills lead to better downstream task adaptation and what metrics can measure these properties.

Our main contributions can be summarized in the following:

1. We theoretically study a novel but practical task adaptation cost (i.e., WAC) for MISL, which measures how well the MISL-learned skills can be applied as downstream task initializations.

2. We propose a novel disentanglement metric (i.e., LSEPIN) that captures both the informativeness and separability of skills. LSEPIN is theoretically related to WAC and can be used to develop URL objectives.

3. We propose a new URL formulation based on Wasserstein distance and extend the above theoretical analysis to it, resulting in novel URL objectives for skill learning. Besides also promoting separability, they could discover more skills than existing MISL that are potentially optimal for downstream tasks.

Although our contribution is mainly theoretical, in appendices H and I we show the feasibility of practical algorithm design with our proposed metrics and empirical examples to validate our results. A summary of our proposed metrics and algorithm is in appendix A and frequently asked questions are answered in appendix B.

## 2 PRELIMINARIES

We consider infinite-horizon MDPs $\mathcal{M} = (\mathcal{S}, \mathcal{A}, P, p_0, \gamma)$ *without external rewards* with discrete states $\mathcal{S}$ and actions $\mathcal{A}$, dynamics $P(s_{t+1}|, s_t, a_t)$, initial state distribution $p_0(s_0)$, and discount factor $\gamma \in [0, 1]$. A policy $\pi(a|s)$ has its discounted state occupancy measure as $p^\pi(s) = (1 - \gamma) \sum_{t=0}^\infty \gamma^t P_t^\pi(s)$, where $P_t^\pi(s)$ is the probability that policy $\pi$ visits state **s** at time $t$. There can be downstream tasks that define extrinsic reward as a state-dependent function $r(s)$, where action-dependent reward functions can be handled by modifying the state to include the previous action. The cumulative reward of the corresponding downstream task is $\mathbb{E}_{p^\pi(s)}[r(s)]$.

We formulate the problem of unsupervised skill discovery as learning a skill-conditioned policy $\pi(a_t|s_t, z)$ where $z \in \mathcal{Z}$ represents the latent skill and $\mathcal{Z}$ is a discrete set. $H(\cdot)$ and $I(\cdot; \cdot)$ denote entropy and mutual information, respectively. $W(\cdot, \cdot)$ denotes Wasserstein distance. We use upper-case letters for random variables and lower-case letters for samples, eg. $s \sim p(S)$.

### 2.1 MUTUAL INFORMATION SKILL LEARNING

Unsupervised skill learning algorithms aim to learn a policy $\pi(A|S, Z)$ conditioned on a latent skill $z$. Their optimization objective is usually the mutual information $I(S; Z)$ and they differ on the prior or approximation of this objective (Gregor et al., 2017; Eysenbach et al., 2019; Achiam et al., 2018; Hansen et al., 2020).

In practical algorithms, the policy is generally denoted as $\pi_\theta(A|S, z_{\text{input}})$ with parameters $\theta$ and conditioned on an skill latent $z_{\text{input}} \sim p(Z_{\text{input}})$. Let $p^{\pi_\theta}(S|z_{\text{input}})$ denote the state distribution of policy. The practical objective of MISL could be:

$$\max_{\theta, p(Z_{\text{input}})} I(S; Z_{\text{input}}) = \mathbb{E}_{p(Z_{\text{input}})}[D_{\text{KL}}(p^{\pi_\theta}(S|z_{\text{input}}) \parallel p^{\pi_\theta}(S))], \tag{1}$$

Policy parameters $\theta$ and the latent variable $Z_{\text{input}}$ can be composed into a single representation, $z = (\theta, z_{\text{input}})$, then $\pi_\theta(A|S, z_{\text{input}}) = \pi(A|S, z)$. We call representation $z$ "skill" in the following paper. Then, MISL is learned by finding an optimal $p(Z)$ that solves

$$\max_{p(Z)} I(S; Z) = \mathbb{E}_{p(Z)}[D_{\text{KL}}(p(S|z) \parallel p(S))]., \tag{2}$$

where $p(S) = \mathbb{E}_{p(Z)}[p(S|z)]$, is the average state distribution of discovered skills.

### 2.2 INFORMATION GEOMETRY OF MISL

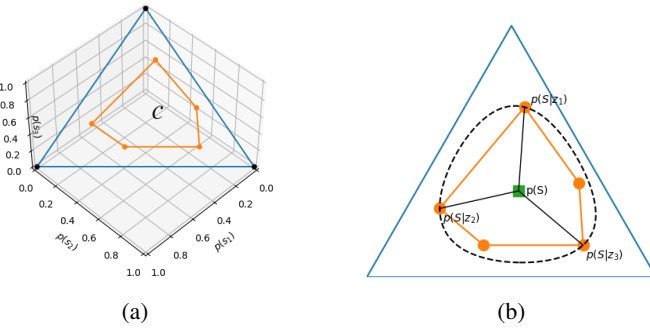

(a)                                                          (b)

Figure 1: Visualized examples: (a) $\mathcal{C}$ is the feasible state distribution set, and the blue simplex is the probability simplex for state distribution with $|\mathcal{S}| = 3$. (b) MISL discovers 3 skills at the vertices $\{z_1, z_2, z_3\}$ on the "circle" with maximum "radius" centered in their average state distribution $p(S)$.

Prior work Eysenbach et al. (2022) shows that the set $\mathcal{C}$ of state distributions feasible under the dynamics of the MDP constitutes a convex polytope lying on a probability simplex of state distributions, where every point in the polytope $\mathcal{C}$ is represented by a skill latent $z$ and its state distribution is $p(S|z)$. For any downstream task defined by a reward function $r : \mathcal{S} \to \mathcal{R}$, because of the linearity of $\mathbb{E}_{p(s)}[r(s)]$ and convexity of $\mathcal{C}$, the state distribution that maximizes the cumulative reward $\mathbb{E}_{p(s)}[r(s)]$

lies at one of the vertices of $\mathcal{C}$ (Boyd & Vandenberghe, 2014). Equation (2) shows that MISL learns a skill distribution $z \sim p(Z)$ to put weight on skills that have maximum KL divergence to the average state distribution. It can be considered as finding skills that lie on the unique (uniqueness proved in appendix E) "circle" with maximum "radius" inside the polytope $\mathcal{C}$, thus the discovered skills lie at the vertices of polytope $\mathcal{C}$, as shown in Lemma 6.5 of Eysenbach et al. (2022) by Theorem 13.11 of Cover & Thomas (2006). So the skills discovered by MISL are optimal for some downstream tasks. An intuitive example of the skills discovered by MISL is shown in fig. 1b.

## 3 THEORETICAL RESULTS

Although MISL discovers some vertices that are potentially optimal for certain downstream tasks, when the downstream task favors target state distributions at the undiscovered vertices, which often happens in practice that the learned skills are not optimal for downstream tasks, there exists a "distance" from discovered vertices to the target vertex, and the "distance" from the initial skill for adaptation to the target state distribution can be considered as the adaptation cost. The prior work only analyzes the adaptation cost from the average state distribution of skills $p(S) = \mathbb{E}_z[p(S|z)]$ to the target state distribution. Because most practical MISL algorithms initialize the adaptation procedure from one of the learned skills (Lee et al., 2019; Eysenbach et al., 2019; Liu & Abbeel, 2021b; Laskin et al., 2021) instead of the average $p(S)$, the prior analysis provides little insight on why these practical algorithms work. The fundamental question for unsupervised skill learning remains unanswered: How the learned skills can be used for downstream task adaptation and what properties of the learned skills are desired for better downstream task adaption?

We have answered this question with theoretical analysis in this section, empirical validation of the theories is in appendix I. Our informal results are as follows:

1. In order to have a low adaptation cost when initializing from one of the learned skills, the learned skills need to be diverse and separate from each other. Separability means the discriminability between states inferred by different skills.

2. MISL alone does not necessarily guarantee diversity and separability. We propose a disentanglement metric LSEPIN to complement MISL for diverse and separable skills.

3. MISL discovers limited vertices, we propose WSEP metric based on Wasserstein distance that can promote diversity and separability as well as discover more vertices than MISL. One Wasserstein distance-based algorithm PWSEP can even discover all vertices.

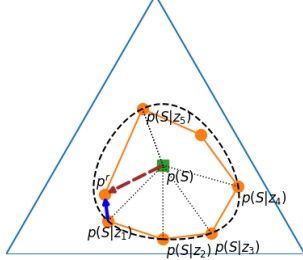

Figure 2: Example of concyclic vertices:

$z_1, z_2, z_3, z_4, z_5$ can all be vertices optimal for MISL, $p(S)$ can be identified by $|\mathcal{S}| = 3$ of them, so eq. (2) can be solved by any 3 of these vertices. The blue arrow is to adapt from learned skills to target distribution for downstream task (our setting) and the brown arrow is to adapt from the average state distribution (setting in Eysenbach et al. (2022))

The first point is intuitive that the diverse and separable skills are likely to cover more potentially useful skills, as shown by empirical results in Eysenbach et al. (2019); Park et al. (2022b); Laskin et al. (2022). The second point claims MISL alone does not guarantee diversity and separability, and this can be seen from the example in fig. 2. In this case, there are two sets of $|\mathcal{S}| = 3$ skills $\mathcal{Z}_a : \{z_1, z_4, z_5\}$ and $\mathcal{Z}_b : \{z_2, z_3, z_5\}$ both on the maximum "circle" solving MISL. Because $z_2$ and $z_3$ have close state distributions, skills of $\mathcal{Z}_b$ are less diverse and less separable. There can be more than $|S|$ vertices on the maximum "circle" in the case of fig. 2 because, unlike prior work Eysenbach et al. (2022), we do not take into account the "non-concyclic" assumption that limits the number of vertices on the same "circle" to be $|S|$. Our proposed disentanglement metric LSEPIN would favor $\mathcal{Z}_a$ over $\mathcal{Z}_b$, and theoretical analysis of LSEPIN is conducted in section 3.2 to show its relation to downstream task adaptation cost. The downstream task procedure we consider is initialized from one of the learned tasks, for the case in fig. 2, when the target state distribution is $p^r$, we consider adapting from the skill in $\mathcal{Z}_a$ that is closest to $p^r$ (blue arrow), which is $z_1$, while the prior work adapts from $p(S)$ (brown arrow).

The advantages of Wasserstein distance are that it is a true metric satisfying symmetry and triangle inequality. We can use it to measure distances that can not be measured by KL divergences. Optimizing these distances also promotes diversity and separability as well as results in better vertex discovery, even capable of discovering all vertices and solving the open question of "vertex discovery" from Eysenbach et al. (2022). Details about Wasserstein distance skill learning are shown in section 3.3. A summary of all proposed metrics and algorithm is in appendix A.

## 3.1 How to measure diversity and separatability of learned skills

Many previous MISL algorithms (Eysenbach et al., 2019; Gregor et al., 2017; Sharma et al., 2020) emphasized the importance of diversity and tried to promote diversity by using uniform $p(Z_{\text{input}})$ for eq. (1). However, uniform $p(Z_{\text{input}})$ for objective eq. (1) does not ensure diverse $z$ for $p(Z)$ in eq. (2) since $z = (\theta, z_{\text{input}})$ also depends on the learned parameter $\theta$. We show an example in appendix D when maximizing $I(S; Z)$ with uniform $p(Z_{\text{input}})$ results in inseparable skills. Empirical discussions in Park et al. (2022b); Laskin et al. (2022) also mentioned that the learned skills of these MISL methods often lack enough diversity and separability. Furthermore, as mentioned previously by the example in fig. 2, even when $I(S; Z)$ in eq. (2) is maximized, the learned skills could still lack diversity and separability of the skills. To complement MISL, we propose a novel metric to explicitly measure the diversity and separability of learned skills.

We consider $I(S; \mathbf{1}_z)$ ($\mathbf{1}_z$ is the binary indicator function of $Z = z$) to measure the informativeness and separability of an individual skill $z$. In the context of unsupervised skill learning, informativeness should refer to the information shared between a skill and its inferred states. As mentioned, separability means the states inferred by different skills should be discriminable. We analyze the minimum of $I(S; \mathbf{1}_z)$ over learned skills. We name it Least SEParability and INformativeness (LSEPIN)

$$\text{LSEPIN} = \min_z I(S; \mathbf{1}_z). \tag{3}$$

$I(S; \mathbf{1}_z)$ is related to how much states inferred by skill $z$ and states not inferred by $z$ are discriminable from each other, so it covers not only informativeness but also separability of skills. In the context of representation learning, the metrics capturing informativeness and separability are called the disentanglement metrics (Do & Tran, 2019b; Kim et al., 2021), so we also call LSEPIN as a disentanglement metric for unsupervised skill learning. More details about the difference between disentanglement for representation learning and disentanglement for our skill learning setting are in appendix F.

## 3.2 How disentanglement affects downstream task adaptation

We provide a theoretical justification for the proposed disentanglement metric, showing that it can be a complement of $I(S; Z)$ to evaluate how well the URL agent is prepared for downstream tasks by the following theorems.

**Definition 3.1** (Worst-case Adaptation Cost). Worst-case Adaptation Cost (WAC) is defined as

$$\text{WAC} = \max_r \min_{z \in \mathcal{Z}^*} D_{\text{KL}}(p(S|z) \,\|\, p^r), \tag{4}$$

where $p^r$ is the optimal feasible state marginal distribution for the downstream task defined by $r$, and $\mathcal{Z}^*$ is the set of learned skills.

The following theoretical results show how the LSEPIN metric is related to the WAC in definition 3.1.

**Theorem 3.1.** *When learned skill sets $\mathcal{Z}_i, i = 1, 2, ...$ with $N \leq |\mathcal{S}|$ skills ($N$ skills have $p(z) > 0$) sharing the same skill $z$ are all MISL solutions, The skill set with the higher $I(S; \mathbf{1}_z)$ will have higher $p(z)$ and lower adaptation cost for all $r_z$ in the set $\mathcal{R}_z$, where $\mathcal{R}_z$ is the set of downstream tasks always satisfying $\forall i, \forall r \in \mathcal{R}_z$, $z = \arg\max_{z' \in \mathcal{Z}_i} D_{\text{KL}}(p(S|z') \,\|\, p^{r_z})$. And the maximum of this adaptation cost has the following formulation:*

$$IC_z = \max_{r \in \mathcal{R}_z} \frac{C_z(r) - p(z)D_z(r)}{1 - p(z)}, \tag{5}$$

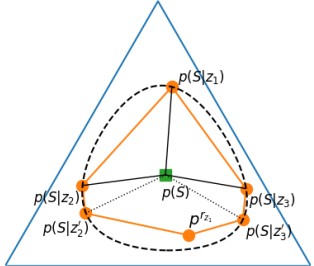

the MISL objective $I(S; Z)$ is maximized by solutions

Figure 3: Example of two MISL solutions: $\mathcal{Z}_1^* : \{z_1, z_2, z_3\}$ and $\mathcal{Z}_2^* : \{z_1, z_2', z_3'\}$, $\mathcal{Z}_2^*$ has higher $I(S; \mathbf{1}_{z_1})$

*where*

$$C_z(r) = I(S; Z) + D_{\text{KL}}(p(S) \parallel p^r), \tag{6}$$
$$D_z(r) = D_{\text{KL}}(p(S|z) \parallel p^r). \tag{7}$$

Theorem 3.1 provides a correlation between our proposed metric $I(S; \mathbf{1}_z)$ the adaptation cost $\text{IC}_z$ from a skill in $\mathcal{Z} \setminus \{z\}$ that is closest to the downstream task optimal distribution and its detailed proof is in appendix C.1. To better understand the claim of this theorem, we can look at the intuitive example shown in fig. 3. In this case $|\mathcal{S}| = 3$. When MISL is maximized by 3 skills, the skill combinations as MISL solutions could be $\mathcal{Z}_1^* : \{z_1, z_2, z_3\}$ and $\mathcal{Z}_2^* : \{z_1, z_2', z_3'\}$. $\mathcal{Z}_2^*$ has higher $I(S; \mathbf{1}_{z_1})$ than $\mathcal{Z}_1^*$. By theorem 3.1, solution $\mathcal{Z}_2^*$ should have lower cost to adapt to the optimal distribution $p^{r_{z_1}}$ of the downstream task $r_{z_1}$.

**Corollary 3.1.1.** *When the MISL objective $I(S, Z)$ is maximized by $N \leq |\mathcal{S}|$ skills, WAC is bounded of a solution $\mathcal{Z}^*$ by*

$$WAC \leq \max_{z \in \mathcal{Z}^*} IC_z = \max_{z \in \mathcal{Z}^*} \max_{r \in \mathcal{R}_z} \frac{C_z(r) - p(z)D_z(r)}{1 - p(z)}. \tag{8}$$

*WAC is the worst-case adaptation cost defined in definition 3.1, $C_z$ and $D_z$ are as defined in eqs. (6) and (7). $\mathcal{R}_z$ here needs to satisfy $\forall r \in \mathcal{R}_z$, $z = \arg\max_{z' \in \mathcal{Z}^*} D_{\text{KL}}(p(S|z') \parallel p^r)$.*

Corollary 3.1.1 provides an upper bound for WAC. The proof is deferred to appendix C.2. The results in Corollary 3.1.1 and theorem 3.1 considered situations when MISL is solved and $I(S; Z)$ is maximized, we also discussed how $I(S; \mathbf{1}_{z_1})$ and LSEPIN affects learned skills and adaptation cost when $I(S; Z)$ is not maximized in appendix C.4.

By theorem 3.1 we know that higher $I(S; \mathbf{1}_z)$ implies lower $\text{IC}_z$, but how much $\text{IC}_z$ associated with an individual skill $z$ contribute to the overall WAC can not be known in prior and it depends on specific $C_z$ and $D_z$. Moreover, specific $C_z$ and $D_z$ depend on the "shape" of the undiscovered parts of $\mathcal{C}$ and can not be known before the discovery of all vertices. Therefore, in practice, like existing work (Durugkar et al., 2021; He et al., 2022) treating the desired properties of each skill equally in practical algorithms, we could treat every $\text{IC}_z$ equally. We have the following theorem showing under which assumptions we can treat every $\text{IC}_z$ equally for WAC.

**Theorem 3.2.** *When 1. the optimal state distribution for the downstream task is far from $p(S)$ and 2. The state space is large, i.e. $|\mathcal{S}|$ is large. $\text{IC}_z$ of all learned skills can be considered equally contribute to WAC.*

Both assumptions for this theorem are practical and can commonly happen in complex and high-dimensional environments. When every $IC_z$ is treated equally for WAC, optimizing LSEPIN could lead to lower WAC. It is formally analyzed and proven in appendix C.3.

In summary, we have provided theoretical insight on how $I(S; \mathbf{1}_z)$ affects downstream task adaptation and how optimizing LSEPIN could lower WAC under practical assumptions. We do not assume "non-concyclic" vertices and we consider the practical approach of directly adapting from learned skills instead of the average state distribution. LSEPIN is a complement to the mutual information objective $I(S; Z)$. Compared to $I(S; Z)$, it provides a better metric to evaluate the effectiveness of learned MISL skills for potential downstream tasks. Our results have shown the diversity and separability of the learned skills measured by $I(S; \mathbf{1}_z)$ and LSEPIN are desired for better downstream task adaptation.

**Remark 3.2.1.** *One limitation with MISL even with LSEPIN is that even without the limitation of the number of skills to have $p(z) > 0$, it still can not discover vertices $v$ such that*

$$D_{\text{KL}}(p(S|v) \parallel p(S)) < \max_{p(z)} \mathbb{E}_{p(z)} \left[ D_{\text{KL}}(p(S|z) \parallel p(S)) \right]$$

*Vertex $p^{r_{z_1}}$ in fig. 3 belongs to such vertices.*

## 3.3 SKILL LEARNING WITH WASSERSTEIN DISTANCE

In this subsection, we analyze a new strategy that replaces the KL divergence in information geometry with Wasserstein distance for better geometric properties to overcome the limitation of MISL shown in remark 3.2.1.

Maximizing $I(S; Z)$ and LSEPIN are essentially maximizing distances measured by KL divergences between points in a polytope. KL divergence is not symmetric and does not satisfy the triangle inequality, so KL divergences between points of the polytope could be incomparable when two KL divergences don't share a same point. We study the strategy that replaces the KL divergence in MISL with Wasserstein distance since Wasserstein distance is a true metric. Then we conduct further theoretical analysis to exploit its better geometrical properties such as symmetry and triangle inequality.

In this section, we will introduce the learning objectives as well as evaluation metrics for **W**asserstein **D**istance **S**kill **L**earning (WDSL), analyze what kind of skills these objectives can learn, where the learned skills lie in the polytope, and how these learned skills contribute to downstream task adaptation. Theoretically, the favored property of WDSL is that it discovers more vertices in $\mathcal{C}$ that are potentially optimal for downstream tasks than MISL, and one WDSL algorithm can discover all vertices.

### 3.3.1 OBJECTIVES FOR WASSERSTEIN DISTANCE SKILL LEARNING

First of all, we can trivially replace the KL divergences in the MISL objective eq. (2) with Wasserstein distance and obtain a basic WDSL objective

$$\max_{p(z)} \mathbb{E}_{p(z)} \left[ W(p(S|z), p(S)) \right]. \tag{9}$$

We name it **A**verage **W**asserstein skill learning **D**istance (AWD), similar to the MISL objective in eq. (2), this objective also learns skills that lie on a hyper ball with a maximum radius. Because this objective is not our main proposition and also suffers from the limitation of remark 3.2.1, we put the analysis of this objective in appendix G.1.

We mainly analyze this objective for WDSL:

$$\text{WSEP} = \sum_{z_i \in \mathcal{Z}} \sum_{z_j \in \mathcal{Z} \setminus \{z_i\}} W(p(S|z_i), p(S|z_j))), \tag{10}$$

where $\mathcal{Z}$ is the set of skills with $p(z) > 0$. We call this objective **W**asserstein **SEP**aratibility (WSEP), it can be considered as a disentanglement for WDSL as it measures the Wasserstein distance between learned skills. Recall that separability for MISL is defined as how discriminable the state is, Wasserstein distances between skills can not only represent discriminability but also can express the distance between trajectories when there are no overlappings.

### 3.3.2 GEOMETRY OF LEARNED SKILLS

As mentioned before in section 2.2, the skills that are potentially optimal for downstream tasks lie at the vertices of the polytope $\mathcal{C}$ of feasible state distributions. By the following lemma, we show that optimizing WSEP will push the learned skills to the vertices of the polytope.

**Lemma 3.3.** *When WSEP is maximized, all learned skills with $p(z) > 0$ must lie at the vertices of the polytope.*

Proof of this lemma is in appendix G.2.

The previous theoretical results of disentanglement metric LSEPIN depend on the maximization of $I(S; Z)$, so as mentioned in remark 3.2.1, it still only discover vertices with maximum "distances" to the average distribution $p(S)$. However, WSEP does not depend on the maximization of other objectives, e.g., eq. (9), so there is no distance restriction on the vertices discovered by WSEP. Therefore, it is possible for WSEP to discover all vertices of the feasible polytope $\mathcal{C}$, thus discovering all optimal skills for potential downstream tasks. For example, in an environment with a polytope shown in fig. 1b, MISL only discovers 3 vertices on the "circle" with maximum "radius" while WSEP is able to discover all 5 vertices.

**Remark 3.3.1.** *When there is no limitation on the number of skills with positive probability, Maximizing WSEP could discover more vertices than MISL in some cases, and even potentially discover all vertices, as shown in the example appendix G.6.*

### 3.3.3 How WSEP affects downstream task adaptation

Then, we propose a theorem about how the WSEP metric is related to downstream task adaptation when there is a limitation on the quantity of learned skills.

**Definition 3.2.** *Mean Adaptation Cost (MAC): mean of the Wasserstein distances between the undiscovered vertices and the learned skills closest to them.*

$$MAC = \frac{1}{|\mathcal{V} \setminus \mathcal{Z}^*|} \sum_{z' \in \mathcal{V} \setminus \mathcal{Z}^*} \min_{z \in \mathcal{Z}^*} W(p(S|z'), p(S|z)) \tag{11}$$

$\mathcal{V}$ *is the set of all skills that have their conditional state distribution at vertices of the MDP's feasible state distribution polytope, and $\mathcal{Z}^*$ is all learned skills with $p(z) > 0$.*

**Theorem 3.4.** *When WSEP is maximized by $|\mathcal{Z}^*|$ skills, the MAC can be upper-bounded:*

$$MAC \leq \frac{\sum_{z \in \mathcal{Z}^*} L_{\mathcal{V}}^z - (|\mathcal{Z}^*| - 1)WSEP}{|\mathcal{V} \setminus \mathcal{Z}^*||\mathcal{Z}^*|} \tag{12}$$

$$MAC \leq \frac{\sum_{z \in \mathcal{Z}^*} L_{\mathcal{V}} - (|\mathcal{Z}^*| - 1)WSEP}{|\mathcal{V} \setminus \mathcal{Z}^*||\mathcal{Z}^*|}, \tag{13}$$

*where*

$$
\begin{aligned}
L_{\mathcal{V}}^z &= \sum_{v \in \mathcal{V}} W(p(S|v), p(S|z)) \\
L_{\mathcal{V}} &= \max_{v' \in \mathcal{V}} \sum_{v \in \mathcal{V}} W(p(S|v), p(S|v'))
\end{aligned}
\tag{14}
$$

Theorem 3.4 shows the relation between WSEP and the upper bounds of adaptation cost MAC in the practical setting, where the number of skills to be learned is limited. The proof is in appendix G.3.

In a stationary MDP, the polytope is fixed, so the edge lengths $L_{\mathcal{V}}^z$ and $L_{\mathcal{V}}$ are constant. Larger WSEP seems to tighten the bounds, but different WSEP also means a different $\mathcal{Z}^*$ set of learned skills, thus a different $\sum_{z \in \mathcal{Z}^*} L_{\mathcal{V}}^z$. Therefore, increasing WSEP only tightens the bound in eq. (13) but not necessarily the bound in eq. (12).

**Remark 3.4.1.** *More distance is not always good: WSEP as a disentanglement metric promotes the distances between learned skills and These two bounds of MAC show that maximizing WSEP can indeed help with downstream task adaptation, but this does not mean that learned skills with more WSEP will always result in lower MAC. An illustrative example is shown in appendix G.5, where more distant skills with higher WSEP do not have lower adaptation costs*

**Remark 3.4.2.** *If we replace the Wasserstein distances in WSEP with KL divergences, we get a symmetric formulation of $KLSEP = \sum_{z_i \in \mathcal{Z}} \sum_{z_j \in \mathcal{Z}, i \neq j} D_{KL}(p(S|z_i) \parallel p(S|z_j))$. It is symmetric, but it does not promote diversity and separability because KL divergence does not satisfy the triangle inequality. More details are analyzed in appendix G.8*

WSEP does not suffer from the limitation of remark 3.2.1 because it does not try to find skills on a maximum "circle". Although WSEP can potentially discover more vertices than MISL, we find that it may not be able to discover all vertices of the feasible state distribution polytope $\mathcal{C}$ in appendix G.7.

### 3.3.4 Solving the vertex discovery problem

The following theorem shows a learning procedure based on Wasserstance distance capable of discovering all vertices of feasible state distribution polytope $\mathcal{C}$.

**Theorem 3.5.** *When $\mathcal{V}$ is the set of all vertices of the feasible state distribution polytope $\mathcal{C}$, all $|\mathcal{V}|$ vertices can be discovered by $|\mathcal{V}|$ iterations of maximizing*

$$PWSEP(i) : \min_{\lambda} W\Big(p(S|z_i), \sum_{z_j \in \mathcal{Z}_i} \lambda^j p(S|z_j)\Big), \tag{15}$$

*where $\mathcal{Z}_i$ is the set of skills discovered from iteration 0 to $i-1$ and $z_i$ is the skill being learned at $i$th iteration. $\lambda$ is a convex coeffcient of dimension $i-1$ that every element $\lambda^j \geq 0, \forall j \in \{0, 1, .., i-1\}$ and $\sum_{j \in \{0,1,..,i-1\}} \lambda^j = 1$.*

*In the initial iteration when $\mathcal{Z}_i = \emptyset$, PWSEP(0) can be $W(p(S|z_0), p(S|z_{rand}))$ with $z_{rand}$ to be a randomly initialized skill.*

PWSEP($i$) can be considered as a projection to the convex hull of $\mathcal{Z}_i$, so we call it Projected WSEP and this learning procedure the PWSEP algorithm. It can discover all $|\mathcal{V}|$ vertices with only $|\mathcal{V}|$ skills. Although lemma 3.3 shows that maximizing WSEP also discovers vertices, the discovered vertices could be duplicated (shown in appendix G.7). Maximizing projected distance PWSEP($i$) could ensure the vertex learned at each new iteration is not discovered before. Proof and more analysis of the vertex discovery problem can be found in appendix G.4.

## 4 RELATED WORK

MISL is widely implemented and has been the backbone of many URL algorithms (Achiam et al., 2018; Florensa et al., 2017; Hansen et al., 2020). Prior work Eysenbach et al. (2022) tried to provide theoretical justification for the empirical prevalence of MISL from an information geometric (Amari & Nagaoka, 2000), but their analysis mainly considered an unpractical downstream task adaptation procedure. Works like Eysenbach et al. (2019); Park et al. (2022b; 2023); He et al. (2022); Laskin et al. (2022) showed the empirical advantages of favored properties such as diversity and separability of learned skills. Our theoretically justified these properties and showed they benefit practical adaptation.

In Kim et al. (2021) the concept of disentanglement was mentioned. They used the SEPIN@k and WSEPIN metrics from representation learning (Do & Tran, 2019b) to promote the informativeness and separability between different dimensions of the skill latent. However, properties of latent representations could be ensured by optimization only in the representation space, so they do not explicitly regulate the state distributions of learned skills like our proposed LSEPIN and WSEP do. Appendix F discussed more details.

Recent practical unsupervised skill learning algorithms (He et al., 2022; Durugkar et al., 2021) maximize a lower bound of WSEP, so our analysis on WSEP provides theoretical insight on why these Wasserstein distance-based unsupervised skill learning algorithms work empirically. Their empirical results showed the feasibility and usefulness of skill discovery with Wasserstein distance.

Successor feature (SF) method SFOLS (Alegre et al., 2022) can also discover all vertices but learns an over-complete set of skills, which our PWSEP algorithm efficiently avoids. In appendix G.4.2, the difference between the SF setting and our skill learning setting is discussed in detail, as well as the comparison of theoretical properties between our proposed PWSEP and SFOLS. Other methods like Hansen et al. (2020); Liu & Abbeel (2021b) combined MISL with SF for URL, and they are shown to accelerate downstream task adaptation. Since they are MISL methods adapting from one of the learned skills, our theoretical results also apply to them.

## 5 CONCLUSION

We investigated the geometry of task adaptation from skills learned by unsupervised reinforcement learning. We proposed a disentanglement metric LSEPIN for mutual information skill learning to capture the diversity and separability of learned skills, which are critical to task adaptation. Unlike the prior analysis, we are able to build a theoretical connection between the metric and the cost of downstream task adaptation. We further proposed a novel strategy that replaces KL divergence with Wasserstein distance and extended the geometric analysis to it, which leads to novel objective WSEP and algorithm PWSEP for unsupervised skill learning. Our theoretical result shows why they should work, what could be done, and what limitations they have. Specifically, we found that optimizing the proposed WSEP objective can discover more optimal policies for potential downstream tasks than previous methods maximizing the mutual information objective $I(S; Z)$. Moreover, the proposed PWSEP algorithm based on Wasserstein distance can theoretically discover all optimal policies for potential downstream tasks.

Our theoretical results could inspire new algorithms using LSEPIN or Wasserstein distance for unsupervised skill learning. For Wasserstein distance, the choice of transport cost is important, which may require strong prior knowledge. Our future work will develop practical algorithms that learn deep representations such that common transport costs such as L2 distance in the representation space can accurately reflect the difficulty of traveling from one state to the other.

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

## A  SUMMARY OF PROPOSED METRICS AND ALGORITHM

Our results are mainly theoretical, showing how properties measured by the following metrics are related to downstream task adaptation. The results can inspire new algorithms with these metrics.

We proposed $I(S; \mathbf{1}_z)$ to measure the separability of an individual skill $z$, more about separability is discussed in appendix F. It is linked to the adaption cost $IC_z$ that initializes from the closest skill other than skill $z$ (in the set $\mathcal{Z} \setminus \{z\}$) to the downstream task optimal distribution.

We proposed LSEPIN $= \min_z I(S; \mathbf{1}_z)$ to measure the overall separability and diversity of learned skills. It is linked to the adaption cost WAC defined in eq. (4) that initializes from the closest skill to the downstream task optimal distribution in the learned skill set $\mathcal{Z}$.

We proposed WSEP $= \sum_{z_i \in \mathcal{Z}} \sum_{z_j \in \mathcal{Z} \setminus \{z_i\}} W(p(S|z_i), p(S|z_j)))$ to measure the overall separability and diversity of learned skills. It is linked to the adaption cost MAC defined in eq. (11) that initializes from the closest skill to the downstream task optimal distribution in the learned skill set $\mathcal{Z}$. WSEP can discover more skills that are potentially optimal for downstream tasks.

We proposed the PWSEP algorithm. Theoretically, it can discover all skills potentially optimal for downstream tasks.

Table 1: Properties of proposed metrics

|  | $I(S; \mathbf{1}_z)$ | **LSEPIN** | **WSEP** |
|---|---|---|---|
| Measurement target | An individual skill | Set of learned skills | Set of learned skills |
| "Distance" metric | KL divergence | KL divergence | Wasserstein distance |
| Related adaptation cost | $IC_z$ | WAC | MAC |
| Practical objective | No | Yes | Yes |

Table 1 displays the summarised properties of the proposed metrics.

## B  FREQUENTLY ASKED QUESTIONS

### B.1  WHAT'S THE OVERLAPS AND DIFFERENCES BETWEEN OUR WORK AND EYSENBACH ET AL. (2022)?

**Differences:** Their analysis of how MISL affects downstream task adaptation is limited to the adaptation procedure of initializing from the average state distribution $p(S) = \mathbb{E}_{z \in \mathcal{Z}}[p(S|z)]$ of learned skill set $\mathcal{Z}$. Because most practical MISL algorithms initialize the adaptation procedure from one of the learned skills (Lee et al., 2019; Eysenbach et al., 2019; Hansen et al., 2020; Liu & Abbeel, 2021b; Laskin et al., 2021) instead of the average $p(S)$, their analysis provides little insight on why these practical algorithms work. Unlike Eysenbach et al. (2022), our work analyzed the practical and popular adaptation procedure of initializing from the learned skills. Our theoretical results not only provide insight on why the above-mentioned practical algorithms work but also answer the fundamental question for unsupervised skill learning: How the learned skills can be used for downstream task adaptation, and what properties of the learned skills are desired for better downstream task adaption? Moreover, our proposed metrics and algorithm are novel and our analysis for the Wasserstein distance offers an innovative perspective on unsupervised skill learning.

**Overlap:** The only overlap between our work and the prior work (Eysenbach et al., 2022) is that we both analyze the geometry of the state distributions of the skills learned by unsupervised skill learning methods.

### B.2  WHAT EXACTLY DOES SEPARABILITY MEAN AND HOW IT IS RELATED TO DIVERSITY?

Separability of a skill means the discriminability between states inferred by different skills. For example, a high $D_{\mathrm{KL}}(p(S|z) \| p(S|Z \neq z))$ or high $W(p(S|z), p(S|Z \neq z))$ could mean skill $z$ is more "distant" from other skills. We show in appendix F that increasing $I(S; \mathbf{1}_z)$ leads to higher $D_{\mathrm{KL}}(p(S|z) \| p(S|Z \neq z))$, thus promoting separability.

Separability is the prerequisite for diversity. Without separability, even with a large number of skills, they could be close to each other and only cover a small region of the feasible state distribution polytope, so skills need to be distinctive and separable first before they can be diverse.

### B.3 What's the difference between $I(S; \mathbf{1}_z)$ and $I(S; Z)$?

**High-level Intuitive difference:**

The binary indicator $\mathbf{1}_z$ is a random variable that takes value in $\{0, 1\}$ with the distribution $p(\mathbf{1}_z)$, which only describes skill $z$, while $Z$ is a random variable with the distribution $p(Z)$, which describes all skills. $I(S; \mathbf{1}_z)$ enables to measure how well every single skill is learned in terms of informativeness and separability, so we can evaluate the minimum $\min_z I(S; \mathbf{1}_z)$ by LSEPIN and the median by MSEPIN(L719); While the default Mutual Information Skill Learning (MISL) objective $I(S; Z)$ only measures the overall informativeness of all skills.

**Formal difference:**

$I(S; \mathbf{1}_z)$ can be decomposed as

$$I(S; \mathbf{1}_z) = p(z)D_{KL}(p(S|z)||S) + p(Z \neq z)D_{KL}(p(S|Z \neq z)||S), \quad (16)$$

while $I(S; Z)$ can be decomposed as

$$I(S; Z) = E_{z \sim p(Z)}[D_{KL}(p(S|z)||S)]. \quad (17)$$

The distribution $p(S|Z \neq z) = E_{z' \sim p(Z' \neq z|z)}[p(S|z')]$ is in the composition of $I(S; \mathbf{1}_z)$ but not in the composition of $I(S; Z)$. Notice that the second term in the for decomposition of $I(S; \mathbf{1}_z)$:

$$D_{KL}(p(S|Z \neq z)||S) \neq E_{z' \sim p(Z' \neq z|z)}[D_{KL}(p(S|z')||S)]. \quad (18)$$

So these are two different formulations.

**Advantages of $I(S; \mathbf{1}_z)$:**

1. $I(S; \mathbf{1}_z)$ can be used to evaluate every single skill.

2. Besides informativeness, $I(S; \mathbf{1}_z)$ explicitly encourages separability, more details about separability are discussed in appendix F.

### B.4 Why not replace the Wasserstein distance in WSEP or PWSEP with KL divergence?

Because KL divergence does not satisfy the triangle inequality, maximizing the replaced WSEP objective could jeopardize the diversity of learned skills as shown in appendix G.8. Appendix G.4 also discussed why KL divergence can not replace the Wasserstein distance in PWSEP.

### B.5 Is it practical to treat the disentanglement of every skill equally?

Yes, theoretically, as the proof of theorem 3.2 in appendix C.3 shows, the contribution of $IC_z$ to WAC can be treated equally for each skill $z$ under two assumptions:

1. The downstream task favored state distribution is far from the average state distribution $p(S)$.

2. The state space is large.

Both could commonly happen in high-dimensional practical environments. When the contribution of $IC_z$ to WAC can be treated equally for each skill $z$, the disentanglement metric $I(S; \mathbf{1}_z)$ for each skill $z$ can be considered equally important.

Practically, existing unsupervised skill learning algorithms such as He et al. (2022); Durugkar et al. (2021) treat the desired properties of every skill equally.

### B.6 Can one infer the shape of $\mathcal{C}$ or perform MAP estimation to get the exact weight of each $IC_z$ for WAC?

It is unnecessary and could be infeasible. As mentioned, it is practical to treat $IC_z$ of each skill $z$ equally important for WAC. By Maximum A Posteriori (MAP) estimation, one still needs a prior of the polytope shape, then update the belief of the shape by the samples of state distributions instead of just state samples. One needs to know which kind of state distribution is feasible for this particular MDP instead of just exploring the state space.

Undiscovered vertices are not within the convex hull of discovered skills, so you can not combine the discovered skills to get an undiscovered state distribution. You will have zero samples of the KL divergence between learned skills and the undiscovered state distributions, on which the MAP update depends.

It might be possible for the specific setting where you can infer the possible state distributions by the explored states. However, the purpose of this estimation is just to obtain a weight about treating some skills more important than others. which can be too expensive for practical algorithm design, and bias or lag in estimation could make it worse than just treating every skill equally.

### B.7 Does $I(S; \mathbf{1}_z)$ and LSEPIN affect downstream task performance when $I(S; Z)$ is not maximized?

Yes, empirically, the importance of diversity and separability of skills are mentioned in many prior works (Eysenbach et al., 2019; Laskin et al., 2022; He et al., 2022; Park et al., 2022b), disentanglement metrics like $I(S; \mathbf{1}_z)$ and LSEPIN can explicitly measure these properties. Theoretically, it is discussed in appendix C.4.

## C Proof for Theoretical results in section 3.2

### C.1 Proof for Theorem 3.1

**Theorem 3.1.** *When learned skill sets $\mathcal{Z}_i, i = 1, 2, ...$ with $N \leq |\mathcal{S}|$ skills ($N$ skills have $p(z) > 0$) sharing the same skill $z$ are all MISL solutions, The skill set with the higher $I(S; \mathbf{1}_z)$ will have higher $p(z)$ and lower adaptation cost for all $r_z$ in the set $\mathcal{R}_z$, where $\mathcal{R}_z$ is the set of downstream tasks always satisfying $\forall i, \forall r \in \mathcal{R}_z$, $z = \arg\max_{z' \in \mathcal{Z}_i} D_{\mathrm{KL}}(p(S|z') \parallel p^{r_z})$. And the maximum of this adaptation cost has the following formulation:*

$$IC_z = \max_{r \in \mathcal{R}_z} \frac{C_z(r) - p(z)D_z(r)}{1 - p(z)}, \tag{5}$$

*where*

$$C_z(r) = I(S; Z) + D_{\mathrm{KL}}(p(S) \parallel p^r), \tag{6}$$
$$D_z(r) = D_{\mathrm{KL}}(p(S|z) \parallel p^r). \tag{7}$$

*Proof.* First, it is not necessary for the skill sets to be all MISL solutions. The proof of this theorem only needs the skill sets to satisfy these **necessary conditions**:

1. All $\mathcal{Z}_i$ share the same average state distribution $p(S) = \mathbb{E}_{z' \in \mathcal{Z}_i}[p(S|z')]$

2. $D_{\mathrm{KL}}(p(S|z') \parallel p(S))$ is constant for all $z'$ in all $\mathcal{Z}_i$.

3. All $\mathcal{Z}_i$ share the same skill $z$

In this proof, we use $z$ to denote the skill shared among the skill sets and $z' \in \mathcal{Z}_i$ to denote a general skill in set $\mathcal{Z}_i$.

A MISL solution is a skill distribution $p(Z)$ with $N$ skills having positive probabilities. Different solutions have the same $p(S) = \mathbb{E}_{z' \in \mathcal{Z}_i} p(S|z')$. Because for MISL solutions, every learned skill $z' \in \mathcal{Z}_i$ should satisfy $p(S|z') = \arg\max_{p \in \mathcal{C}} D_{\mathrm{KL}}(p \parallel p(S))$. It is like the skill-conditioned state distributions of MISL solutions are all on the "largest circle" that is centered in $p(S)$, with the

"radius" being $\max_{p \in \mathcal{C}} D_{\mathrm{KL}}(p \parallel p(S))$ (see Section 6.2 of Eysenbach et al. (2022)). Therefore, MISL solutions sharing the same $z$ satisfy the necessary conditions.

The proof of Lemma 6.3 in Eysenbach et al. (2022) has shown that average state marginal $p(S)$ is a vector in $|\mathcal{S}|$-dimensional space with $|\mathcal{S}| - 1$ degrees of freedom. MISL can recover at most $|\mathcal{S}|$ unique skills because every learned skill $z'$ should satisfy $p(S|z') = \arg\max_{p \in \mathcal{C}} D_{\mathrm{KL}}(p \parallel p(S))$, thus more than $|\mathcal{S}|$ unique skills put more than $|\mathcal{S}|$ distance constraints on $p(S)$, making it overly specified and ill-defined. Using a fixed number of skills is common in practice, so we use $N \leq |\mathcal{S}|$ skills for the MISL problem.

Because we do not consider the assumption in Eysenbach et al. (2022) restricting at most $|\mathcal{S}|$ skills on the maximum "circle" to solve MISL, the could be more than $|\mathcal{S}|$ skills on the maximum "circle", so $N$ can be $|\mathcal{S}|$ and different $|\mathcal{Z}_i|$ can all have $|\mathcal{S}|$ skills.

The sketch of the proof goes like the following:

1. The sum of KL divergences from learned skills to the optimal target distribution for a downstream task is constant.

2. The upper bound of $IC_z$ depends this constant and $p(z)$, and the upper bound decreases monotonically with higher $p(z)$.

3. Increasing $I(S; \mathbf{1}_z)$ results in higher $p(z)$ thus lower upper bound of $IC_z$.

We can see that for a solution $\mathcal{Z}_i$ the weighted sum of the KL divergence between the optimal vertex of considered downstream task $r \in \mathcal{R}_z$ and the MISL learned skills is:

$$\sum_{z' \in \mathcal{Z}_i} p(z') D_{\mathrm{KL}}(p(S|z') \parallel p^r) \tag{19}$$

$$= \sum_{z' \in \mathcal{Z}_i} p(z') \mathbb{E}_{p(S|z')}[\log \frac{p(S|z')}{p^r}]$$

$$= \sum_{z' \in \mathcal{Z}_i} \sum_S p(S, z')[\log \frac{p(S|z')}{p^r}] \tag{20}$$

$$= -H(S|Z) + H(p(S), p^r) \tag{21}$$

$$= H(S) - H(S|Z) - H(S) + H(p(S), p^r) \tag{22}$$

$$= I(S; Z) + D_{\mathrm{KL}}(p(S) \parallel p^r), \tag{23}$$

where $H(p(S), p^r)$ denotes the cross-entropy of the $p^r$ relative to $p(S)$.

Because $I(S; Z) = \mathbb{E}_{z' \in \mathcal{Z}_i}[D_{\mathrm{KL}}(p(S|z') \parallel p(S))]$ and $p(S)$ are constant under the necessary conditions 1 and 2, the sum only depends on $r \in \mathcal{R}_z$ and we can consider

$$\sum_{z' \in \mathcal{Z}_i} p(z) D_{\mathrm{KL}}(p(S|z') \parallel p^r) = C_z(r), \ \forall i \tag{24}$$

Because the minimum is less than the mean, we have

$$(1 - p(z)) \min_{z' \in \mathcal{Z}_i, z' \neq z} D_{\mathrm{KL}}(p(S|z') \parallel p^r) \leq$$
$$\sum_{z' \in \mathcal{Z}_i, z' \neq z} p(z') D_{\mathrm{KL}}(p(S|z') \parallel p^r), \ \forall i \tag{25}$$

then

$$\min_{z' \in \mathcal{Z}_i, z' \neq z} D_{\mathrm{KL}}(p(S|z') \parallel p^r) \leq \frac{C_z(r) - p(z) D_{\mathrm{KL}}(p(S|z) \parallel p^r)}{1 - p(z)}, \ \forall i \tag{26}$$

We have assumed $z = \arg\max_z D_{\mathrm{KL}}(p(S|z) \parallel p^r)$, so $\arg\min_{z'} D_{\mathrm{KL}}(p(S|z') \parallel p^r)$ is chosen from $z' \in \mathcal{Z}_i, z' \neq z$, so

$$\min_{z' \in \mathcal{Z}_i} D_{\mathrm{KL}}(p(S|z') \parallel p^r) \leq \frac{C_z(r) - p(z) D_z(r)}{1 - p(z)}, \ \forall i \tag{27}$$

and we can bound $\min_{z'} D_{\text{KL}}(p(S|z') \| p^r)$ by eq. (27) and obtain the formulation of eq. (5).

For any given downstream task $r \in \mathcal{R}_z$, different MISL solutions $\mathcal{Z}_i$ have the same $C_z(r)$ and $D_z(r)$ but different $p(z)$, so next we show how a higher $I(S; \mathbf{1}_z)$ affects $p(z)$ and then results in a tighter upper bound of $\min_{z'} D_{\text{KL}}(p(S|z') \| p^r)$.

Because

$$
\begin{aligned}
I(S; \mathbf{1}_z) = & p(z) D_{\text{KL}}(p(S|z) \| p(S))) \\
& + (1 - p(z)) D_{\text{KL}}(p(S|Z \neq z) \| p(S)),
\end{aligned}
\tag{28}
$$

and

$$
p(S|Z \neq z) = \frac{p(S) - p(z)p(S|z)}{1 - p(z)},
\tag{29}
$$

for the same skill $z$, the value of $I(S; \mathbf{1}_z)$ could only be changed by $p(z)$. If $\frac{\partial I(S; \mathbf{1}_z)}{\partial p(z)}$ is non-negative, a higher $I(S; \mathbf{1}_z)$ would indicate a higher $p(z)$.

$$
\begin{aligned}
\frac{\partial I(S; \mathbf{1}_z)}{\partial p(z)} = & (1 - p(z)) \frac{\partial D_{\text{KL}}(p(S|Z \neq z) \| p(S))}{\partial p(z)} \\
& + D_{\text{KL}}(p(S|z) \| p(S))) - D_{\text{KL}}(p(S|Z \neq z) \| p(S))
\end{aligned}
\tag{30}
$$

By the necessary condition 2 and by convexity of KL divergence, because $p(S|Z \neq z)$ is a convex combination of skills not equal to $z$, we have

$$
D_{\text{KL}}(p(S|z) \| p(S)) \geq D_{\text{KL}}(p(S|Z \neq z) \| p(S))
\tag{31}
$$

. then $\frac{\partial I(S; \mathbf{1}_z)}{\partial p(z)}$ is non-negative if

$$
\frac{\partial D_{\text{KL}}(p(S|Z \neq z) \| p(S))}{\partial p(z)} \geq 0.
\tag{32}
$$

By eq. (29), we have:

$$
p'(S|Z \neq z) = \frac{p(S) - p'(z)p(S|z)}{1 - p'(z)}
\tag{33}
$$

where $p'(z) = p(z) + \Delta$, and $\Delta$ is a small positive value that keeps the changed $p'(Z)$ satisfying the necessary conditions described at the beginning of the proof. Then

$$
p(S|Z \neq z) = \lambda p'(S|Z \neq z) + (1 - \lambda)p(S)
\tag{34}
$$

$$
\lambda = \frac{1 - p'(z)}{1 - p(z)} \frac{p(z)}{p'(z)}
\tag{35}
$$

So $0 < \lambda < 1$. By convexity of KL divergence, we have

$$
\begin{aligned}
& D_{\text{KL}}(p(S|Z \neq z) \| p(S)) & (36) \\
= & D_{\text{KL}}(\lambda p'(S|Z \neq z) + (1 - \lambda)p(S) \| p(S)) & (37) \\
\leq & \lambda D_{\text{KL}}(p'(S|Z \neq z) \| p(S)) + (1 - \lambda) \cdot 0 & (38) \\
< & D_{\text{KL}}(p'(S|Z \neq z) \| p(S)) & (39)
\end{aligned}
$$

and

$$
\frac{\partial D_{\text{KL}}(p(S|Z \neq z) \| p(S))}{\partial p(z)}
\tag{40}
$$

$$
= \lim_{\Delta \to 0^+} \frac{D_{\text{KL}}(p'(S|Z \neq z) \| p(S))}{\Delta}
$$

$$
- \frac{D_{\text{KL}}(p(S|Z \neq z) \| p(S))}{\Delta}
\tag{41}
$$

$$
\geq 0
\tag{42}
$$

Therefore, $\frac{\partial I(S;\mathbf{1}_z)}{\partial p(z)}$ is non-negative, a higher $I(S;\mathbf{1}_z)$ would indicate a higher $p(z)$.

With this result, we go back to eq. (27). Because $D_{\mathrm{KL}}(p(S|z) \parallel p^r)$ is required to be $\max_{z'\in\mathcal{Z}_i} D_{\mathrm{KL}}(p(S|z') \parallel p^r), \forall i, \forall r \in \mathcal{R}_z$, we have

$$D_{\mathrm{KL}}(p(S|z) \parallel p^r) = D_z(r) \geq \sum_{z'\in\mathcal{Z}_i} p(z')D_{\mathrm{KL}}(p(S|z') \parallel p^r) = C_z(r), \ \forall i, \forall r \in \mathcal{R}_z \qquad (43)$$

and because of eq. (43)

$$\frac{\partial \frac{C_z(r)-p(z)D_z(r)}{1-p(z)}}{\partial p(z)} \leq 0, \forall r \in \mathcal{R}_z \qquad (44)$$

Now we have proved that a higher $I(S;\mathbf{1}_z)$ indicates a higher $p(z)$, thus a tighter upper bound in eq. (27) for all $r \in \mathcal{R}_z$, including the worst-case defined in eq. (5). The theorem is then proved. $\qquad\square$

### C.2 PROOF OF COROLLARY 3.1.1

**Corollary 3.1.1.** *When the MISL objective $I(S, Z)$ is maximized by $N \leq |\mathcal{S}|$ skills, WAC is bounded of a solution $\mathcal{Z}^*$ by*

$$WAC \leq \max_{z\in\mathcal{Z}^*} IC_z = \max_{z\in\mathcal{Z}^*} \max_{r\in\mathcal{R}_z} \frac{C_z(r) - p(z)D_z(r)}{1-p(z)}. \qquad (8)$$

*WAC is the worst-case adaptation cost defined in definition 3.1, $C_z$ and $D_z$ are as defined in eqs. (6) and (7). $\mathcal{R}_z$ here needs to satisfy $\forall r \in \mathcal{R}_z$, $z = \arg\max_{z'\in\mathcal{Z}^*} D_{\mathrm{KL}}(p(S|z') \parallel p^r)$.*

*Proof.* The formulation can be directly obtained from eq. (27) and definition 3.1.

The remaining problem here is whether $\bigcup_{z'\in\mathcal{Z}^*} \mathcal{R}_z$ contains all possible downstream tasks.

If there exists a downstream task $r$ not in $\bigcup_{z'\in\mathcal{Z}^*} \mathcal{R}_z$, then this $r$ should satisfy that $\arg\max_{z'\in\mathcal{Z}^*} D_{\mathrm{KL}}(p(S|z') \parallel p^{r_z})$ is not in $Z^*$, which is contradictory. Therefore $\bigcup_{z'\in\mathcal{Z}^*} \mathcal{R}_z$ contains all possible downstream tasks and this corollary is proved. $\qquad\square$

### C.3 DIRECT CORRELATION BETWEEN WAC AND LSEPIN

By theorem 3.1 we know that increasing $I(S;\mathbf{1}_z)$ tightens the upper bound of the adaptation cost $IC_z$ associated with an individual skill $z$. However, how much $IC_z$ of an individual $z$ contributes to the overall WAC depends on the specific $C_z$ and $D_z$ of each skill $z$, and $C_z$ and $D_z$ cannot be known in prior. Therefore, $I(S;\mathbf{1}_z)$ of certain skills could be more important than others for WAC, which can only be known when we already discovered all vertices of the polytope $\mathcal{C}$.

Without prior knowledge, we could only treat every $IC_z$ equally and assume that the cost associated with individual skills would have the same importance on the WAC. The following results provide assumptions that are sufficient to treat the cost of individual skills equally and directly associate LSEPIN with WAC under these assumptions, which require:

- The optimal state distribution for the downstream task to be far away from $p(S)$ (eq. (46))
- The state space is large (eq. (45))

Both requirements are common in practice.

**Lemma C.1.** *When the MISL objective $I(S; Z)$ is maximized by $N \leq |\mathcal{S}|$ skills, the solution set of skills is $\mathcal{Z}^*$, assume that $\forall z \in \mathcal{Z}^*$,*

$$D_m = \min_{r\in\mathcal{R}_z} D_{\mathrm{KL}}(p(S|z) \parallel p^r) \qquad (45)$$

$$C = D_{\mathrm{KL}}(p(S) \parallel p^r), \ \forall r \in \mathcal{R}_z \qquad (46)$$

*where $C$ and $D_m$ are two constants, $\mathcal{R}_z$ is the set of reward functions that satisfy $\forall r \in \mathcal{R}_z$, $z = \arg\max_{z'\in\mathcal{Z}^*} D_{\mathrm{KL}}(p(S|z') \parallel p^{r_z})$. Then, higher LSEPIN results in lower WAC.*

*Proof.* We denote

$$D_z(r) = D_{\mathrm{KL}}(p(S|z) \parallel p^r) \tag{47}$$

$$C_z(r) = \sum_{z'} p(z') D_{\mathrm{KL}}(p(S|z') \parallel p^r) \tag{48}$$

First, by eqs. (23) and (46), we have

$$C_z(r) = -H(S|Z) + H(p(S), p^r) \tag{49}$$

$$= H(S) - H(S|Z) + D_{\mathrm{KL}}(p(S) \parallel p^r) \tag{50}$$

$$= I(S;Z) + D_{\mathrm{KL}}(p(S) \parallel p^r) \tag{51}$$

$$= I(S;Z) + C = C_m \tag{52}$$

$C_m$ is a constant because for MISL solutions, $I(S;Z)$ is maximized and is a constant.

Because $0 \leq p(z) \leq 1$, we have, $\forall 0 \leq p(z) \leq 1$:

$$\arg\max_{r \in \mathcal{R}_z} \frac{C_z(r) - p(z)D_z(r)}{1 - p(z)} = \arg\max_{r \in \mathcal{R}_z} \frac{C_m - p(z)D_z(r)}{1 - p(z)} = \arg\min_{r \in \mathcal{R}_z} D_z(r) \tag{53}$$

Combine eqs. (8), (45) and (53), we have

$$WAC \leq \max_{z \in \mathcal{Z}^*} \frac{C_m - p(z)D_m}{1 - p(z)} \tag{54}$$

Because $C_m$ and $D_m$ is constant across different $z$, and $D_m \geq C_m$ by eqs. (45), (46) and (48), increasing $\min_{z \in \mathcal{Z}^*} p(z)$ decreases right hand side of eq. (54). By eqs. (28) and (29) and non-negativity of $\frac{\partial I(S; \mathbf{1}_z)}{\partial p(z)}$, increasing LSEPIN could lead to higher $\min_{z \in \mathcal{Z}^*} p(z)$ thus lower WAC. $\quad\square$

**Theorem 3.2.** *When 1. the optimal state distribution for the downstream task is far from $p(S)$ and 2. The state space is large, i.e. $|\mathcal{S}|$ is large. $IC_z$ of all learned skills can be considered equally contribute to WAC.*

*Proof.* Lemma C.1 shows the direct correlation between WAC and LSEPIN under assumptions in eq. (46) and eq. (45).

$C$ in Eq. 46 could assume the feasible optimal state distribution for downstream task $r_z$ to have maximum "distance" from the average distribution $p(S)$. It is common in practice that the optimal state distributions for downstream tasks are "far" from the average state distribution $p(S)$ of MISL solutions. Eq. 45 assumes a constant $D_m$, this can apply to practical situations when $|\mathcal{S}|$ is large.

By definition of $R_z$ in theorem 3.1, we can see

$$\min_{r \in R_z} D_{KL}(p(S|z)||p^r) = \max_{z' \neq z} \max_{r \in R_z} D_{KL}(p(S|z')||p^r) = D_m(z) \tag{55}$$

let $\mathcal{Z}'$ denote the set containing learned skills that maximizes $\max_{z' \neq z} \max_{r \in R_z} D_{KL}(p(S|z')||p^r)$ and $z' \neq z$. By eq. (48), we have

$$D_m(z) \leq \frac{C_m}{p(z) + \sum_{z' \in \mathcal{Z}'} p(z')} \tag{56}$$

Because eq. (46) assumed $p^r$ to be on a "maximum circle" centered in $p(S)$, $p^{r^*}$ should have $|\mathcal{S}| - 2$ degrees of freedom, where $r^*$ solves $\min_{r \in R_z} D_{KL}(p(S|z)||p^r)$. Including $z$, there should be $|\mathcal{S}| - 1$ skills that satisfies $D_{KL}(p(S|z')||p^{r^*}) = D_m(z)$ to determine $p^{r^*}$ and $D_m(z)$, which together have $|\mathcal{S}| - 1$ degrees of freedom. So we have $|\mathcal{Z}'| = |\mathcal{S}| - 2$ excluding $z$, and

$$D_m(z) \leq \frac{C_m}{p(z) + \sum_{z' \in \mathcal{Z}'} p(z')} = \frac{C_m}{1 - \sum_{z'' \in \mathcal{Z}^* \setminus \mathcal{Z}' \setminus \{z\}} p(z'')} \tag{57}$$

Without loss of generality, we consider situations when $|\mathcal{Z}^*| = |\mathcal{S}|$, when $\mathcal{Z}^* \setminus \mathcal{Z}' \setminus \{z\}$ contains only one skill. We denote it $z''$, then:

$$D_m(z) \leq \frac{C_m}{1 - p(z'')} \tag{58}$$

Because optimizing LSEPIN increases $p(z)$ for all $z \in \mathcal{Z}^*$, the larger $|\mathcal{S}|$ is, the tighter the bound in eq. (58) is. And because $D_m(z) \geq C_m$ by definition, when $|\mathcal{S}|$ is large, all $D_m(z)$ can be considered as constant close to $C_m$.

Therefore, combining the above analysis on the assumptions of eqs. (45) and (46) and lemma C.1, we proved the claim of theorem 3.2. □

### C.4 How $I(S; \mathbf{1}_z)$ affects the learned skills when MISL is not optimized

As mentioned by the proof of theorem 3.1 in appendix C.1, the claim that higher $I(S; \mathbf{1}_z)$ leads to lower $IC_z$ applies also to any skills sets satisfying the necessary conditions:

1. All $\mathcal{Z}_i$ share a common skill $z$ and the same average state distribution $p(S) = \mathbb{E}_{z' \in \mathcal{Z}_i}[p(S|z')]$

2. $D_{\mathrm{KL}}(p(S|z') \parallel p(S))$ is constant for all $z'$ in all $\mathcal{Z}_i$.

In practice, there could be a gap between optimal $\max_{p(Z)} \mathbb{E}_{p(Z)} D_{\mathrm{KL}}(p(S|z) \parallel p(S))$ and a learned suboptimal $d_m = \max_i \mathbb{E}_{z \in \mathcal{Z}_i} D_{\mathrm{KL}}(p(S|z) \parallel \mathbb{E}_{z' \in \mathcal{Z}_i}[p(S|z')])$, where $\mathcal{Z}_i$ are the skill sets that can be learned. This gap could be a result of a limitation in the skill number $|\mathcal{Z}_i|$ or of an optimization error.

In order to maximize $I(S; Z) = \mathbb{E}_{p(Z)} D_{\mathrm{KL}}(p(S|z) \parallel p(S))$ The learned $p(Z)$ would be positive only for the skills that have $D_{\mathrm{KL}}(p(S|z) \parallel p(S)) = d_m$. For two skill sets $\mathcal{Z}_1, \mathcal{Z}_2$ maximizing $I(S; Z)$ to the same suboptimal value $d_m$, they satisfy the necessary condition 2 since every skill has $D_{\mathrm{KL}}(p(S|z) \parallel p(S)) = d_m$.

Exploration is important when $I(S; Z)$ is suboptimal as empirically shown in Laskin et al. (2021), If $\mathcal{Z}_1, \mathcal{Z}_2$ satisfy the necessary condition 1 and share the same $p(S)$, they would have the same exploration of the state space.

Therefore, our theoretical results in theorems 3.1 and 3.2 also apply to practical situations when MISL is not optimally solved, and they showed that when two sets have the same exploration while maximizing $I(S; Z)$ to the same degree, the set with better LSEPIN is preferred for downstream task adaptation.

## D Example when MISL with uniform $p(Z)$ is not guaranteed to promote diversity

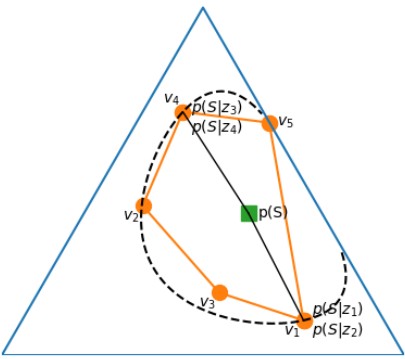

Figure 4: Example MISL with uniform $p(Z)$ does not promote diversity

For a case of with $|S| = 3$, the feasible polytope $\mathcal{C}$ allows a maximum "circle" centered at $[0.2, 0.4, 0.4]$ with a maximum "radius" ($I(S, Z)$) of 0.253 as this fig in shows. There are 5 vertices $v_{1:5}$ of $\mathcal{C}$, of which $v_1, v_2, v_4, v_5$ lie on the maximum "circle". Vertex $v_1$ and vertex $v_4$ are $[0.2, 0.7, 0.1]$ and $[0.2, 0.1, 0.7]$.

Because $\theta$ is shared by all skills and $z = \{\theta, Z_{input}\}$, we can assume $p(Z) = p(Z_{input})$

When there are 4 skills to learn, Uniform $p(Z_{input})$ should have $p(z_{input}) = p(z) = 0.25$ for all $z$. For both uniform $p(Z)$ and maximization of $I(S; Z)$, the optimal skill set $\{z_1, z_2, z_3, z_4\}$ should be the one containing two skills $z_1, z_2$ at $v_1$ and the other two skills $z_3, z_4$ at $v_4$, as shown in the figure. Only with this skill set, the uniform average of skills $p(S) = \sum_z p(z)p(S|z)$ could be the center of the maximum "circle" $[0.2, 0.4, 0.4]$.

We can see from this example that although skill set $\{z_1, z_2, z_3, z_4\}$ maximizes both $I(S; Z)$ and $H(Z_{input})$, there are two pairs of skills being not separable thus resulting in limited diversity

In appendix F, we show higher $I(S; \mathbf{1}_z)$ explicitly increases $D_{KL}(p(S|Z \neq z)\|p(S|z))$ thus promoting $p(S|z)$ to be separable from $p(S|Z \neq z)$ (average state distribution of skills other than $z$). Therefore, LSEPIN promotes diverse skills explicitly. Comparing to skills only at $v_1$ and $v_4$, MISL with LSEPIN would favor skills at $v_1, v_2, v_4, v_5$ respectively. In order to keep $p(S)$ to be the center of maximum "circle" $[0.2, 0.4, 0.4]$, $p(Z)$ of skills at $v_1, v_2, v_4, v_5$ is not necessarily uniform.

# E  UNIQUENESS OF THE "CIRCLE" LEARNED FROM MISL

Maximizing $I(S; Z)$ leads to unique average state distribution $p(S)$, and it can be proven by the following:

1. By Lemma 6.5 of Eysenbach et al. (2022), MISL objective is equivalent to $\min_{p(S)} \max_z D_{KL}(p(S|z) \| p(S))$.

2. Because of the strict convexity of KL divergence, $\max_z D_{KL}(p(S|z) \| p(S))$ is also strictly convex.

3. $p(S)$ is constrained in a convex polytope $\mathcal{C}$.

Therefore it is a convex optimization problem with a strictly convex objective function. The solution is unique. With unique $p(S)$ to be the "center", the "circle" of maximum "radius" $D_{KL}(p(S|z) \| p(S))$ is also unique.

# F  DIFFERENCE BETWEEN DISENTANGLEMENT FOR REPRESENTATION LEARNING AND SKILL LEARNING SETTING

Kim et al. (2021) adopted metrics SEPIN@k and WSEPIN (Do & Tran, 2019a) from representation learning to measure the disentanglement of the skill latent. They focused on the informativeness and separability (independencies) between different dimensions of the representation.

SEPIN@k is the top k average of $I(S, Z_i|Z_{\neq i})$ while WSEPIN is the average of $I(S, Z_i|Z_{\neq i})$ weighed on $I(S, Z_i)$, where $Z_i$ denotes $i$th dimension of the skill latent.

However, by $I(S, Z_i|Z_{\neq i})$ they care about how to efficiently represent the skill with the latent representation rather than promoting the separability and diversity of the trajectories induced by different skills. It can be maximized by only optimization in the latent space regardless of the state distributions in the state space. Therefore, this metric does not directly encourage diversity and separability of state distributions of skills thus not contributing to adaptation costs.

If we directly adopt $I(S, Z_i|Z_{\neq i})$ for our setting, $Z$ should be a one-hot vector. We have

$$I(S, Z_i|Z_{\neq i}) = H(S|Z_{\neq i}) - H(S|Z_i, Z_{\neq i}),$$

Because $Z$ is a one-hot vector, $Z_i$ is completely dependent on $Z_{\neq i}$, i.e. knowing $Z_{\neq i}$ also means knowing $Z_i$. Therefore, $H(S|Z_i, Z_{\neq i}) = H(S|Z_{\neq i})$ and $I(S, Z_i|Z_{\neq i})$ is constantly zero and we can not directly use the disentanglement metrics composed of $I(S, Z_i|Z_{\neq i})$ to explicitly measure the diversity and separability of learned skills.

Instead, we propose to use $I(S; \mathbf{1}_z)$ to measure the informativeness and separability of each individual skill $z$, where $\mathbf{1}_z$ is the binary indicator for $Z = z$. It is a mutual information metric intrinsically measuring informativeness. We will show that it also measures separability.

In the URL skill learning setting, separability between one skill and others entails that the states inferred by this skill should share little overlap with the states inferred by others. Therefore, it means that the states should be certain to be inferred by one single skill, thus meaning low $H(\mathbf{1}_z|S)$.

Because $I(S; \mathbf{1}_z) = H(S) - H(S|\mathbf{1}_z)$, for situations with the same state and skill distributions, the one with larger $I(S; \mathbf{1}_z)$ would be lower in $H(S|\mathbf{1}_z)$. since $H(\mathbf{1}_z|S) = H(S|\mathbf{1}_z) + H(\mathbf{1}_z) - H(S)$, lower $H(S|\mathbf{1}_z)$ means also low in $H(\mathbf{1}_z|S)$ thus better separability.

Moreover, a larger $I(S; \mathbf{1}_z)$ explicity encourages a larger $D_{\mathrm{KL}}(p(S|Z \neq z) \parallel p(S|z))$.

By the proof of eq. (40) in Appendix B.1, we have shown when $I(S; Z)$ is fixed a higher $I(S; \mathbf{1}_z)$ results in a higher $D_{\mathrm{KL}}(p(S|Z \neq z) \parallel p(S))$.

By convexity of KL divergence:

$$D_{\mathrm{KL}}(p(S|Z \neq z) \parallel p(S)) \leq p(z)D_{\mathrm{KL}}(p(S|Z \neq z) \parallel p(S|z)) + \\ (1 - p(z))D_{\mathrm{KL}}(p(S|Z \neq z) \parallel p(S|Z \neq z)) \tag{59}$$

$$\leq p(z)D_{\mathrm{KL}}(p(S|Z \neq z) \parallel p(S|z)) + 0 \tag{60}$$

$$< p(z)D_{\mathrm{KL}}(p(S|Z \neq z) \parallel p(S|z)) \tag{61}$$

So increasing $I(S; \mathbf{1}_z)$ also increases the lower bound of $D_{\mathrm{KL}}(p(S|Z \neq z) \parallel p(S|z))$, thus explicitly encouraging skills to be separated from each other. Therefore, we compose metrics from $I(S; \mathbf{1}_z)$ to measure the disentanglement of learned skills.

## G    PROOF FOR SKILL LEARNING WITH WASSERSTEIN DISTANCES

When using KL divergence, solving eq. (2) learns skills that lie at the vertices of the polytope. This is actually because of the strict convexity KL divergence not mentioned by prior work (Eysenbach et al., 2022), which leads to

$$D_{\mathrm{KL}}(\lambda p_1 + (1 - \lambda)p_2 \parallel p(S)) < \\ \lambda D_{\mathrm{KL}}(p_1 \parallel p(S)) + (1 - \lambda)D_{\mathrm{KL}}(p_2 \parallel p(S)), \tag{62}$$

where $0 < \lambda < 1$ and $p1 \neq p2$. Therefore, the skills at the vertices of the convex polytope always have higher KL divergence from the average state distribution than other skills within the polytope or on the faces of the polytope, because those skills are just convex combinations of the skills at the vertices.

In general, Wasserstein distance $W(\cdot, p)$ is commonly considered to be convex (Santambrogio, 2015; Peyré & Cuturi, 2019). For the 2-Wasserstein distance, its square $W_2^2(\cdot, p)$ considered strictly convex by (2.12) of  Carlen & Gangbo (2003).  In practice, for image observations, similar to Liu & Abbeel (2021a;b) that apply L2 norm-based particle based-entropy (Singh et al., 2003b) in the representation space, we can also apply the 2-Wasserstein distance-based metrics to the representations of observations. So it is practical to consider strictly convex Wasserstein distances for our analysis.

### G.1    ANALYSIS ON THE BASIC WDSL OBJECTIVE

We can trivially replace the KL divergences in the MISL objective eq. (2) with Wasserstein distance and obtain a basic WDSL objective

$$\max_{p(z)} \mathbb{E}_{p(z)}\left[W(p(S|z), p(S))\right]. \tag{63}$$

Similar to Lemma 6.5 of Eysenbach et al. (2022), optimizing eq. (9) is equivalent to solving

$$\min_{p(S)} \max_{p(z)} W(p(S|z), p(S)). \tag{64}$$

We name it **Average Wasserstein skill learning Distance (AWD)**. Therefore, when the adaptation cost for downstream tasks is defined as the Wasserstein distance between the state distributions of the initialization policy and the optimal policy since maximizing AWD learns an average state distribution that is close to worst case policy, this average state distribution will be a worst-case robust initialization.

We have the following Lemma:

**Lemma G.1.** *When AWD is maximized, all learned skills with $p(z) > 0$ must lie at the vertices of the polytope.*

*Proof.* If a skill is not at a vertex of the polytope, it can be represented by a strictly convex combination of some vertices, and for any skill $z$ not at a vertex,

$$W(p(S), p(S|z)) < \sum_{z' \in \mathcal{V}_z} \lambda_{z'} W(p(S), p(S|z')) \leq \max_{z' \in \mathcal{V}_z} W(p(S), p(S|z')), \tag{65}$$

where $\mathcal{V}_z$ is the smallest set of vertices such that $z$ is within the convex hull of them, namely $p(S|z) = \sum_{z' \in \mathcal{V}_z} \lambda_{z'} p(S|z')$, $0 < \lambda_{z'} < 1$ for all $z' \in \mathcal{V}_z$ and $\sum_{z' \in \mathcal{V}_z} \lambda_{z'} = 1$. The first inequality is by strict convexity, and the second is because the maximum should be not less than the mean.

We can see when $p(S|z)$ is not at one of the vertices, there must be a vertex with a higher distance to $p(S)$. Therefore, just like MISL, WDSL with AWD also discovers skills at the vertices. □

Although optimizing AWD discovers vertices like MISL, it also suffers from the limitation of remark 3.2.1 because it also only discovers vertices at the maximum "circle" with maximum Wasserstein distance to the average.

## G.2 PROOF FOR LEMMA 3.3

**Lemma 3.3.** *When WSEP is maximized, all learned skills with $p(z) > 0$ must lie at the vertices of the polytope.*

*Proof.* Similar to the proof of lemma G.1, for any skill $z_i$, when $p(S|z_i)$ is not at the vertices, it can be represented by a strictly convex combination of some vertices, so for all $z_i \in \mathcal{Z}$,

$$\sum_{z_j \in \mathcal{Z} \setminus \{z_i\}} W(p(S|z_i), p(S|z_j))) < \sum_{z' \in \mathcal{V}_{z_i}} \lambda_{z'} \sum_{z_j \in \mathcal{Z} \setminus \{z_i\}} W(p(S|z'), p(S|z_j))$$
$$\leq \max_{z' \in \mathcal{V}_{z_i}} \sum_{z_j \in \mathcal{Z} \setminus \{z_i\}} W(p(S|z'), p(S|z_j)), \tag{66}$$

where $\mathcal{V}_{z_i}$ is the smallest set of vertices such that $z_i$ is within the convex hull of them, namely $p(S|z_i) = \sum_{z' \in \mathcal{V}_{z_i}} \lambda_{z'} p(S|z')$, $0 < \lambda_{z'} < 1$ for all $z' \in \mathcal{V}_{z_i}$ and $\sum_{z' \in \mathcal{V}_{z_i}} \lambda_{z'} = 1$.

When $p(S|z_i)$ is not at one of the vertices, there must be a vertex with a higher $\sum_{z_j \in \mathcal{Z} \setminus \{z_i\}} W(p(S|z_i), p(S|z_j)))$ for all $z_i \in \mathcal{Z}$. Therefore, WSEP is not maximized when there is still a skill not at the vertices. □

## G.3 PROOF FOR THEOREM 3.4

**Theorem 3.4.** *When WSEP is maximized by $|\mathcal{Z}^*|$ skills, the MAC can be upper-bounded:*

$$MAC \leq \frac{\sum_{z \in \mathcal{Z}^*} L_{\mathcal{V}}^z - (|\mathcal{Z}^*| - 1)WSEP}{|\mathcal{V} \setminus \mathcal{Z}^*||\mathcal{Z}^*|} \tag{12}$$

$$MAC \leq \frac{\sum_{z \in \mathcal{Z}^*} L_{\mathcal{V}} - (|\mathcal{Z}^*| - 1)WSEP}{|\mathcal{V} \setminus \mathcal{Z}^*||\mathcal{Z}^*|}, \tag{13}$$

*where*

$$L_{\mathcal{V}}^z = \sum_{v \in \mathcal{V}} W(p(S|v), p(S|z))$$
$$L_{\mathcal{V}} = \max_{v' \in \mathcal{V}} \sum_{v \in \mathcal{V}} W(p(S|v), p(S|v')) \tag{14}$$

*Proof.* By definition,

$$MAC \leq \frac{\sum_{t \in \mathcal{V} \setminus \mathcal{Z}} \sum_{z \in \mathcal{Z}} W(p(S|t), p(S|z))}{|\mathcal{V} \setminus \mathcal{Z}||\mathcal{Z}|}. \tag{67}$$

And we can see

$$\sum_{t \in \mathcal{V} \setminus \mathcal{Z}} \sum_{z \in \mathcal{Z}} W(p(S|t), p(S|z))$$

$$= \sum_{z \in \mathcal{Z}} L_{\mathcal{V}}^z - \sum_{z \in \mathcal{Z}} \sum_{z' \in \mathcal{Z}} W(p(S|z'), p(S|z)) \tag{68}$$

$$= \sum_{z \in \mathcal{Z}} L_{\mathcal{V}}^z - \text{WSEP},$$

where the inequality in eq. (67) is by convexity.

Then the first inequality eq. (12) is proved by combining eq. (67) and eq. (68). Because by definition, for all $z \in \mathcal{Z}$, $L_{\mathcal{V}}^z \leq L_{\mathcal{V}}$, the second inequality eq. (13) also holds. $\qquad \square$

### G.4 PROOF FOR THEOREM 3.5 AND THE VERTEX DISCOVERY PROBLEM

#### G.4.1 PROOF AND ALGORITHM FOR THEOREM 3.5

**Theorem 3.5.** *When $\mathcal{V}$ is the set of all vertices of the feasible state distribution polytope $\mathcal{C}$, all $|\mathcal{V}|$ vertices can be discovered by $|\mathcal{V}|$ iterations of maximizing*

$$PWSEP(i) : \min_{\lambda} W\Big(p(S|z_i), \sum_{z_j \in \mathcal{Z}_i} \lambda^j p(S|z_j)\Big), \tag{15}$$

*where $\mathcal{Z}_i$ is the set of skills discovered from iteration 0 to $i-1$ and $z_i$ is the skill being learned at $i$th iteration. $\lambda$ is a convex coeffcient of dimension $i-1$ that every element $\lambda^j \geq 0, \forall j \in \{0, 1, .., i-1\}$ and $\sum_{j \in \{0,1,..,i-1\}} \lambda^j = 1$.*

*In the initial iteration when $\mathcal{Z}_i = \emptyset$, $PWSEP(0)$ can be $W(p(S|z_0), p(S|z_{rand}))$ with $z_{rand}$ to be a randomly initialized skill.*

*Proof.* We first see that for the initial iteration, by eq. (65), maximizing $W(p(S|z_0), p(S|z_{rand}))$ learns a $z_0$ at one of the vertices.

Then in later iterations when $\mathcal{Z}_i$ is not empty, we need to prove that maximizing PWSEP($i$) can learn a new vertex that was not discovered before.

For every skill $z_i'$ not at the vertices, it can be represented by a strict convex combination of $p(S|z_i') = \sum_{v \in \mathcal{V}_{z_i'}} \lambda_{z_i'}^v p(S|v)$, where $\mathcal{V}_{z_i'}$ is the smallest set of vertices such that $z_i$ is within the convex hull of them, $0 < \lambda_{z_i'}^v < 1$ for all $v \in \mathcal{V}_{z_i'}$ and $\sum_{v \in \mathcal{V}_{z_i'}} \lambda_{z_i'}^v = 1$.

We have

$$\min_{\lambda} W\Big(p(S|z_i'), \sum_{z_j \in \mathcal{Z}_i} \lambda^j p(S|z_j)\Big) = \min_{\lambda} W\Big(\sum_{v \in \mathcal{V}_{z_i'}} \lambda_{z_i'}^v p(S|v), \sum_{z_j \in \mathcal{Z}_i} \lambda^j p(S|z_j)\Big)$$

$$\leq W\Big(\sum_{v \in \mathcal{V}_{z_i'}} \lambda_{z_i'}^v p(S|v), \sum_{v \in \mathcal{V}_{z_i'}} \lambda_{z_i'}^v \sum_{z_j \in \mathcal{Z}_i} \lambda_v^j p(S|z_j)\Big), \tag{69}$$

where

$$\lambda_v = \arg\min_{\lambda} W\Big(p(S|v), \sum_{z_j \in \mathcal{Z}_i} \lambda^j p(S|z_j)\Big). \tag{70}$$

Furthermore, we can see

$$W\Big(\sum_{v \in \mathcal{V}_{z_i'}} \lambda_{z_i'}^v p(S|v), \sum_{v \in \mathcal{V}_{z_i'}} \lambda_{z_i'}^v \sum_{z_j \in \mathcal{Z}_i} \lambda_v^j p(S|z_j)\Big) \leq \sum_{v \in \mathcal{V}_{z_i'}} \lambda_{z_i'}^v W\Big(p(S|v), \sum_{z_j \in \mathcal{Z}_i} \lambda_v^j p(S|z_j)\Big) \tag{71}$$

$$= \sum_{v \in \mathcal{V}_{z_i'}} \lambda_{z_i'}^v \min_{\lambda} W\Big(p(S|v), \sum_{z_j \in \mathcal{Z}_i} \lambda^j p(S|z_j)\Big) \tag{72}$$

Now we have shown, for every skill $z_i'$ not at the vertices, when the maximum vertex is unique,

$$\min_\lambda W\Big(p(S|z_i'), \sum_{z_j \in \mathcal{Z}_i} \lambda^j p(S|z_j)\Big) \leq \sum_{v \in \mathcal{V}_{z_i'}} \lambda_{z_i'}^v \min_\lambda W\Big(p(S|v), \sum_{z_j \in \mathcal{Z}_i} \lambda^j p(S|z_j)\Big) \quad (73)$$

$$< \max_{v \in \mathcal{V}_{z_i'}} \min_\lambda W\Big(p(S|v), \sum_{z_j \in \mathcal{Z}_i} \lambda^j p(S|z_j)\Big) \quad (74)$$

Equation (74) is strict when the vertex to maximize the projected distance is unique. For extreme cases where a face of the convex hull of $\mathcal{Z}_i$ and a face of the polytope $\mathcal{C}$ are "parallel", there could be multiple vertices that maximize PWSEP(i) and make it possible for all points on the "parallel" face of $\mathcal{C}$ also to be optimal. However, this can be easily circumvented in practice by adding the skill being learned $z_i'$ into the $\mathcal{Z}_i$ set for a temporary set of $\mathcal{Z}_i'$ and optimizing the projected distance based on $\mathcal{Z}_i'$ instead of $\mathcal{Z}_i$ once $z_i'$ reaches the optimal "parallel" face of $\mathcal{C}$. By doing so, $\mathcal{Z}_i'$ contains a point $z_i'$ on the "parallel" face of $\mathcal{C}$, and $z_i'$ will have 0 projected distance to the convex hull of the new temporary set $\mathcal{Z}_i'$, so the points on the "parallel" face of $\mathcal{C}$ will not be all optimal for $\mathcal{Z}_i'$. Then, we have excluded an extreme case of non-vertex maxima and we can continue to update $z_i'$ to a vertex without reinitializing a new policy.

Therefore, maximizing PWSEP($i$) in every iteration learns a skill that lies at a vertex. The remaining part is to prove that the vertex discovered after every new iteration will be new and different from the vertices discovered earlier.

If the newly discovered vertex $z_i$ is not new and has been previously discovered, it should be in $\mathcal{Z}_i$, and PWSEP($i$) would be 0.

An undiscovered new vertex $v$ at iteration $i$ should not be in the convex hull of $\mathcal{Z}_i$: If the convex hull of $\mathcal{Z}_i$ contains $v$, the vertex $v$ should be either one of the points in set $\mathcal{Z}_i$ or a strict convex combination of points in $\mathcal{Z}_i$. Since the polytope $\mathcal{C}$ is convex, its vertices cannot be a strict convex combination of other points in $\mathcal{C}$, the vertex $v$ can only be in the set of $\mathcal{Z}_i$ as an already discovered vertex if it is in the convex hull of $\mathcal{Z}_i$.

Therefore, when the set $\mathcal{Z}_i$ does not contain $v_i$, $v_i$ should not be in the convex hull of $\mathcal{Z}_i$, resulting

$$\min_\lambda W\Big(p(S|v_i), \sum_{z_j \in \mathcal{Z}_i} \lambda^j p(S|z_j)\Big) > 0 \quad (75)$$

Then, if there still exists undiscovered new vertices, optimal PWSEP($i$) would be larger than 0, and optimizing PWSEP($i$) finds a new vertex at each new iteration.

In conclusion, PWSEP discovers a new vertex at each new iteration, so it can discover all $|\mathcal{V}|$ vertices with $|\mathcal{V}|$ iterations of policy learning. □

Other statistical distances like total variation distance and Heilinger distance are also true metrics satisfying symmetry, and triangle inequality, but they are not strictly convex. KL divergence is strictly convex, and the information projection $p^* = \arg\min_{p \in P} D_{\mathrm{KL}}(p \parallel q)$ has a unique solution for the convex $P$ set, so it seems that the Wasserstein distance projection in the PWSEP algorithm can be replaced by information projection. however, when the new skill $z_i$ does not cover all the states in the support of distributions in $\mathcal{Z}_i$, which could happen a lot in practice, this projected KL divergence would be infinite and no longer convex in the union of their supports. For the moment projection, which is the reverse information projection $p^* = \arg\min_{p \in P} D_{\mathrm{KL}}(q \parallel p)$, it would also be infinite whenever the new skill $z_i$ discovers new states that are not in the support of distributions in $\mathcal{Z}_i$.

Notice that, the sum of the projected Wasserstein distances

$$\mathrm{SPWD} = \sum_{z_m \in \mathcal{Z}} \min_\lambda W\Big(p(S|z_m), \sum_{z_n \in \mathcal{Z} \setminus \{z_m\}} \lambda^n p(S|z_n)\Big) \quad (76)$$

is not an evaluation metric for skill learning. This means that a higher value of SPWD does not mean better skills for downstream task adaptation. An extreme example is that for $|\mathcal{S}| = 3$ and the polytope $\mathcal{C}$ is a Chiliagon with $|\mathcal{V}| = 1000$ vertices. Then the optimal skill set covering all 1000 vertices would have an SPWD near zero. For another skill set with 999 skills at the same vertex and 1 skill at another

vertex with maximum distance from the prior one, then it is possible that this set could have a greater SPWD than the optimal skill set and this skill set is clearly not optimal since it only covers 2 vertices.

Therefore, unlike $I(S; Z)$, LSEPIN, and WSEP, SPWD is not an evaluation metric and can not be used as an optimization objective for unsupervised skill learning. To discover all vertices, the projected Wasserstein distance needs to be optimized iteratively as the PWSEP algorithm.

---

**Algorithm 1** PWSEP algorithm

---

Initialize a random point $\pi_{rand}$, a real value $v > 0$
$i \leftarrow 0$
$\pi_{z_i} \leftarrow \arg\max_{\pi_{z_i}} W(p(S|z_i), p^{\pi_{rand}})$          ▷ Initial iteration
$\Pi_1 \leftarrow \{\pi_{z_i}\}$
**while** v>0 **do**
     $i \leftarrow i + 1$
     $\pi_{z_i} \leftarrow \arg\max_{\pi_{z_i}} \text{PWSEP}(i)$          ▷ eq. (15)
     $\Pi_{i+1} \leftarrow \Pi_i \cup \{\pi_{z_i}\}$
     $v \leftarrow \max_{z_i} \text{PWSEP}(i)$
**end while**

---

Algorithm 1 is the algorithm described in theorem 3.5, its practical implementation is discussed in appendix H.3 and its empirical performance is discussed in appendix I.

### G.4.2 ABOUT VERTEX DISCOVERY AND SUCCESSOR FEATURE METHOD

To the best of our knowledge, the first method to solve the vertex discovery problem with a finite number of skills is the successor feature-based method SFOLS (Alegre et al., 2022).

SFOLS is designed to find a CCS set defined in Eq.(22) from Appendix.A.5 in their paper. When written in our notation, it would be

$$\text{CCS} = \{\rho^{\pi_z} | \exists w, s.t. \rho^{\pi_z} w \geq \rho^{\pi'_z} w\}, \tag{77}$$

where $\rho^{\pi_z}$ is the occupancy measure of skill $z$, $w$ is equivalent to the reward function of states $r$. This definition of CCS also applies to any skill that lies at the edges or faces of the polytope, not necessarily at the vertices; therefore, SFOLS might need to learn a skill number that is much larger than the vertex number $|\mathcal{V}|$ before discovering all vertices. Even for their more rigorous definition in their Eq.(6), where they constrained the skills to be in the non-dominated multi-objective set $\mathcal{F}$, it still does not exclude skills at the edges or faces, because vertices do not Pareto dominate the points at one of its edges, i.e. there always exist tasks on which the vertices have fewer accumulative rewards than points at their edges.

Therefore, SFOLS might need a number of skills $|\mathcal{Z}|$ greater than the number of vertices $|\mathcal{V}|$ to discover all vertices and solve the vertex discovery problem, while PWSEP is guaranteed to discover all $|\mathcal{V}|$ vertices with only $|\mathcal{V}|$ skills.

To contain only vertices, a set modified from CCS should be:

$$\{\rho^{\pi_z} | \exists \mathcal{W}, s.t. |\mathcal{W}| \geq |\mathcal{S}| - 1, \text{ its elements are linearly independent and } \forall w \in \mathcal{W}, \rho^{\pi_z} w \geq \rho^{\pi'_z} w\}, \tag{78}$$

A vertex should be optimal for any task that has optimal solutions on its adjacent edges or faces, so a vertex should be optimal for at least $|\mathcal{S}| - 1$ linearly independent $w$. This could inspire future successor feature algorithms to learn more general policies that lie at the vertices instead of faces of polytope $\mathcal{C}$.

Moreover, there are fundamental differences between our setting of skill discovery and the setting of successor feature methods. For unsupervised skill learning with MISL or Wasserstein distance, the latent representation for skills is only used for indexing, its exact value does not affect the state distributions of the learned skills; While for the successor feature setting, the value of the weight directly determines its associated state distribution, which is the reason why optimization in the weight space is required for the successor feature methods.

That is to say, for an ideal unsupervised skill learning algorithm like PWSEP, it should be able to discover $|V|$ vertices with a skill space containing only $|V|$ points. This is impossible for successor feature learning that needs a weight (used as skills) space $R^{|S|}$ (Appendix.A.5 of Alegre et al. (2022)) containing infinite points.

However, successor feature methods can be considered directly solving eq. (2) instead of eq. (1). Because each weight $w$ determines an optimal parameter $\theta(w)$ for its optimal policy $\pi_{\theta(w)}$. Equation (1) for successor feature setting becomes:

$$
\begin{aligned}
\max_{\theta, p(W)} I(S; W) &= \mathbb{E}_{p(W)}[D_{\mathrm{KL}}(p(S|w, \theta) \| p(S, \theta))] \\
&= \mathbb{E}_{p(W)}[D_{\mathrm{KL}}(p(S|w, \theta(w)) \| \mathbb{E}_{p(W)}[p(S|w, \theta(w))])] \\
&= \mathbb{E}_{p(W)}[D_{\mathrm{KL}}(p(S|w) \| p(S))]
\end{aligned}
\tag{79}
$$

Therefore, methods that combine successor features with MISL such as Hansen et al. (2020); Liu & Abbeel (2021b) are the practical algorithms closer to the setting (optimizing eq. (2)) of our theoretical analysis.

### G.5    EXAMPLE WHEN HIGHER WSEP RESULTS IN HIGHER MAC

In fig. 5. It shows a polytope with 4 vertices in a 3-dimensional space. Suppose in a situation where two skills are learned, and the learned skills are at vertices $a$ and $c$. This situation would be optimal for maximizing WSEP, its MAC is the mean of the length of edge $cd$ and edge $cb$, and its worst-case adaptation cost is the length of edge $cb$. Then for another situation where two skills lie at vertices $a$ and $d$, it has a lower WSEP, but its MAC is the mean of the length of edge $cd$ and edge $db$, which is lower than the former situation. Its worst-case adaptation cost is the length of edge $cd$, which is also lower. In this case, $L_{\mathcal{V}}^d < L_{\mathcal{V}}^c$, so from skill combination $a$ and $c$ to skill combination $a$ and $d$, although WSEP decreased, $\sum_{z \in \mathcal{Z}} L_{\mathcal{V}}^z - WSEP$ also decreased due to the change of $L_{\mathcal{V}}^z$, resulting in tighter upper bound in eq. (12).

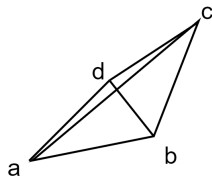

Figure 5: A polytope in a 3-dimensional space, there are 4 vertices.

For a practical example, this situation can be an environment with a state space of one-dimensional integers. If you have two skills, for maximum WSEP, they would put their probability mass close to $-\infty$ and $\infty$, which could be infinitely far away from any sampled downstream task optimal distribution.

### G.6    AN MDP EXAMPLE WHEN WSEP DISCOVERS MORE VERTICES THAN MISL

Consider an MDP with $|\mathcal{S}| = 3$ for states and $|\mathcal{A}| = 3$ for actions.

The transition matrix for action $a_1$ is:

$$
\begin{pmatrix}
0.2 & 0.7 & 0.1 \\
0.2 & 0.7 & 0.1 \\
0.2 & 0.7 & 0.1
\end{pmatrix}
$$

The transition matrix for action $a_2$ is:

$$
\begin{pmatrix}
0.2 & 0.1 & 0.7 \\
0.2 & 0.1 & 0.7 \\
0.2 & 0.1 & 0.7
\end{pmatrix}
$$

The transition matrix for action $a_3$ is:

$$
\begin{pmatrix}
0.4 & 0.3 & 0.3 \\
0.4 & 0.3 & 0.3 \\
0.4 & 0.3 & 0.3
\end{pmatrix}
$$

Because the transition probabilities only depend on actions, state distribution $p(S)$ is determined by distribution $p(A)$:

$$p(S) = p(a_1)[0.2, 0.7, 0.1] + p(a_2)[0.2, 0.1, 0.7] + p(a_3)[0.4, 0.3, 0.3] \tag{80}$$

We can see $p(A)$, in this case, is the convex coefficient of three probabilities, so the feasible state distribution is in the convex polytope $\mathcal{C}$ with these three probabilities being the vertices of $\mathcal{C}$.

The polytope $\mathcal{C}$ produced by this MDP can be shown as the figure below:

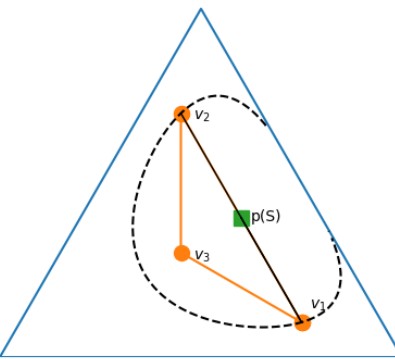

Figure 6: Example when WSEP discovers more vertices than MISL

We label the vertices with $v_1, v_2, v_3$ as shown in fig. 6,

$$\begin{aligned}
p(S|v_1) &= [0.2, 0.7, 0.1] \\
p(S|v_2) &= [0.2, 0.1, 0.7] \\
p(S|v_3) &= [0.4, 0.3, 0.3]
\end{aligned} \tag{81}$$

### G.6.1 MISL ONLY LEARNS SKILLS AT $v_1, v_2$

Firstly, we want to show that the $p(S)$ maximizing $I(S; Z)$ is uniquely determined by the MDP in appendix E. For this specific MDP, the optimal $p(S) = [0.2, 0.4, 0.4]$. To prove $p(S)$ is the solution, we start from the equivalent MISL objective mentioned in appendix E:

$$\min_{p(S)} \max_z D_{\mathrm{KL}}(p(S|z) \parallel p(S)) \tag{82}$$

By strict convexity, for a $p(S)$, the solution $z^*$ of $\max_z D_{\mathrm{KL}}(p(S|z) \parallel p(S))$, must lie at one of the vertices, as shown by eq. (62), also by strict convexity, the value of $\max_z D_{\mathrm{KL}}(p(S|z) \parallel p(S))$ for all $p(S) \neq [0.2, 0.4, 0.4]$ should be strictly larger than its first order Taylor expansion at $[0.2, 0.4, 0.4]$. The gradients of the KL divergences w.r.t. $p(S)$ at $[0.2, 0.4, 0.4]$ are:

$$\begin{aligned}
g_1 &= [-1, -7/4, -1/4] \\
g_2 &= [-1, -1/4, -7/4] \\
g_3 &= [-2, -3/4, -3/4]
\end{aligned} \tag{83}$$

We can represent any other $p(S)$ with $p(S) = \Delta + [0.2, 0.4, 0.4]$, where $\Delta$ can be represented by $[-a - b, a, b]$ ($p(S)$ should be within the probability simplex). By eq. (83), we get the gradients of the KL divergences w.r.t. $[a, b]$ to be:

$$\begin{aligned}
G_1 &= [-0.75, 0.75] \\
G_2 &= [0.75, -0.75] \\
G_3 &= [1.25, 1.25]
\end{aligned} \tag{84}$$

Therefore, we can see when $[a, b] \neq [0, 0]$:

$$\begin{aligned}
\max_z D_{\mathrm{KL}}(p(S|z) \parallel p(S)) > \\
\max(0.253 - 0.75a + 0.75b, 0.253 + 0.75a - 0.75b, 0.105 + 1.25a + 1.25b)
\end{aligned} \tag{85}$$

where 0.253 is Kl divergence from $v_1$ or $v_2$ to $[0.2, 0.4, 0.4]$, and 0.105 is Kl divergence from $v_3$ to $[0.2, 0.4, 0.4]$.

We denote:

$$
\begin{aligned}
f(a,b) &= max(f_1(a,b), f_2(a,b), f_3(a,b)) \\
&= \max(0.253 - 0.75a + 0.75b, 0.253 + 0.75a - 0.75b, 0.105 + 1.25a + 1.25b)
\end{aligned}
\tag{86}
$$

We can see that changing $a, b$ will either increase $f_1$ or $f_2$, because when $[a, b] \neq [0, 0]$:

$$
\max_z D_{\mathrm{KL}}(p(S|z) \parallel p(S)) > f(a,b) \geq max(f_1(a,b), f_2(a,b))
\tag{87}
$$

We can see any $p(S)$ that is not $[0.2, 0.4, 0.4]$ ($[a, b] \neq [0, 0]$) would result in a larger $\max_z D_{\mathrm{KL}}(p(S|z) \parallel p(S))$, so $[0.2, 0.4, 0.4]$ is the only optimal $p(S)$ for maximizing $I(S; Z)$.

MISL would learn only $p(S|z_1), p(S|z_2) = v_1, v_2$ with $p(z_1) = p(z_2) = 0.5$ to have the average state distribution $p(S) = \sum_z (S|z)$ at the unique center $[0.2, 0.4, 0.4]$.

By this solution, $I(S; Z)$ is maximized by $I(S; Z) = \sum_z p(z) D_{KL}(p(S|z) \| p(S)) \approx 0.253$.

Putting weight on the other vertex $z_3$ would lower $I(S; Z)$. For example, if $p(z_1) = p(z_2) = 0.45, p(z_3) = 0.1, p(S|z_3) = v_3$, the average $p(S)$ would be $[0.22, 0.39, 0.39]$ which is not the center of the maximum "circle" any more, and their $I(S; Z) \approx 0.237$.

### G.6.2 WSEP CAN LEARN SKILLS AT ALL VERTICES $v_1, v_2, v_3$

or WSEP, when the skill number is set to $|Z| = 3$, putting 2 skills at $z_1$ and 1 skill at $z_2$ would have lower WSEP than putting 3 skills at $z_1, z_2, z_3$ respectively, because of the triangle inequality. suppose the transportation cost between every two states is 1, then

$W(p(S|z_1), p(S|z_2)) = 0.6$

$W(p(S|z_1), p(S|z_3)) = W(p(S|z_2), p(S|z_3)) = 0.4$

WSEP for 2 skills at $z_1$ and 1 skill at $z_2$ would be 2.4, while WSEP for 3 skills at $z_1, z_2, z_3$ respectively would be 2.8.

Therefore, unlike MISL, WSEP would favor learning all 3 vertices instead of only 2.

### G.7 WSEP CAN NOT ALWAYS DISCOVER ALL VERTICES

For a $|\mathcal{S}| = 4$ case, suppose the polytope $\mathcal{C}$ of feasible state distributions is 3-dimensional and there are 4 vertices $\{v_1, v_2, v_3, v_4\}$, and

$$
\sum_{v \in \{v_1, v_3\}} W(p(S|v), p(S|v_2)) << \sum_{v \in \{v_2, v_3\}} W(p(S|v), p(S|v_1)),
\tag{88}
$$

then if $v_4$ is very close to $v_2$, comparing $v_4$ to a point $v'$ on the surface of $v_1, v_2, v_3$ and very close to $v_1$, it is possible that

$$
\begin{aligned}
&\sum_{v \in \{v_1, v_2, v_3\}} W(p(S|v), p(S|v_4)) \approx \sum_{v \in \{v_1, v_3\}} W(p(S|v), p(S|v_2)) \\
&< \sum_{v \in \{v_1, v_2, v_3\}} W(p(S|v), p(S|v')) \approx \sum_{v \in \{v_2, v_3\}} W(p(S|v), p(S|v_1)).
\end{aligned}
\tag{89}
$$

The skill set of $\{v_1, v_2, v_3, v'\}$ can result in higher WSEP than $\{v_1, v_2, v_3, v_4\}$, so maximizing WSEP can not always discover all vertices.

### G.8 ABOUT WSEP FORMULATION WITH KL DIVERGENCES

Defined in remark 3.4.2

$$
\mathrm{KLSEP} = \sum_{z_i \in \mathcal{Z}} \sum_{z_j \in \mathcal{Z}, i \neq j} D_{\mathrm{KL}}(p(S|z_i) \parallel p(S|z_j))
$$

is a sum of

$$D_{\mathrm{KL}}(p(S|z_i) \parallel p(S|z_j)) + D_{\mathrm{KL}}(p(S|z_j) \parallel p(S|z_i))$$

.

Because KL divergence does not satisfy the triangle inequality, it is possible that

$$
\begin{aligned}
& D_{\mathrm{KL}}(p(S|z_i) \parallel p(S|z_j)) + D_{\mathrm{KL}}(p(S|z_j) \parallel p(S|z_i)) \\
> & D_{\mathrm{KL}}(p(S|z_i) \parallel p(S|z_k)) + D_{\mathrm{KL}}(p(S|z_k) \parallel p(S|z_i)) \\
& + D_{\mathrm{KL}}(p(S|z_i) \parallel p(S|z_k)) + D_{\mathrm{KL}}(p(S|z_k) \parallel p(S|z_i)).
\end{aligned}
\tag{90}
$$

Then learning another skill close to $z_i$ or $z_j$ could result in a higher sum of these KL divergences than learning a new skill $z_k$ that is far from $z_i$ and $z_j$, resulting in poor diversity and separability.

## H    PRACTICAL ALGORITHM DESIGN

Although our main contributions are theoretical analyses, we provide some methods here for approximation of the proposed metrics and show how they can be used for practical algorithm design.

Table 2 shows an overview of methods to approximate the metrics.

Table 2: Approximation of metrics

| Metric | Non-parametric | Parametric |
|--------|----------------|------------|
| $I(S;Z)$ | PBE (Singh et al., 2003a) | $const. + \mathbb{E}[\log p_\phi(z|s)]$ |
| LSEPIN | PBE | $const. + \mathbb{E}[p_\phi(\mathbf{1}_z|S)]$ |
| WSEP | PWD (Rowland et al., 2019) / SWD (Kolouri et al., 2019) | $\mathbb{E}[D_\phi(z,s) - D_\phi(z',s)]$ |

### H.1    PRACTICAL ALGORITHM TO INCORPORATE LSEPIN

For practical algorithms, LSEPIN could play an important role since it measures the informativeness and separability of individual skills and it encourages every learned skill to be distinctive and potentially useful. It is totally feasible to approximate LSEPIN with Particle-Based Entropy (PBE) (Singh et al., 2003a) (details in appendix H.5) or a parametric discriminator that takes state and skill as input $D_\phi(s, z)$. This discriminator can be learned by:

$$\max_\phi \mathbb{E}_{z \sim p(Z)} \left[ \mathbb{E}_{(s,z') \sim p(S|Z \neq z)}[\log(1 - D_\phi(s,z))] + \mathbb{E}_{s \sim p(S|z)}[\log D_\phi(s,z)] \right] \tag{91}$$

Then LSEPIN can be estimated by:

$$\min_{z \in \mathcal{Z}} I(S; \mathbf{1}_z) = H(\mathbf{1}_z) - H(\mathbf{1}_z|S) \tag{92}$$

$$= constant + \mathbb{E}_{s \sim p(S|z)}[\log p(\mathbf{1}_z|S)] \tag{93}$$

$$\approx constant + \mathbb{E}_{s \sim p(S|z)}[\log D_\phi(s,z)] \tag{94}$$

Practical algorithms solve eq. (1) instead of eq. (2), so the desired state distribution of skills $p^{\pi_\theta}(S|Z_{\mathrm{input}})$ is determined by the both $p(Z_{\mathrm{input}})$ and learned $\theta$. In order to let every input skill $z_{\mathrm{input}}$ have equal steps of roll-out, the input skill distribution $p(Z_{\mathrm{input}})$ is often set as a uniform prior (Eysenbach et al., 2019; Sharma et al., 2020). Therefore, $H(\mathbf{1}_z)$ can be considered as a constant. Then the intrinsic reward to increase LSEPIN can be $log D_\phi(s, z)$ for states inferred by skill $z$, and $log(1 - D_\phi(s, z))$ for states not inferred by skill $z$.

### H.2    PRACTICAL ALGORITHM FOR WSEP

How to efficiently compute the Wasserstein Distances between states of different skills is fundamental for WSEP. In He et al. (2022), approaches like Sliced Wasserstein Distances (SWD) (Kolouri et al., 2019) and Projected Wasserstein distance (PWD) (Rowland et al., 2019) have been demonstrated useful for unsupervised skill discovery. Here we propose an alternative approach to inspire future

algorithm design. We can parametrize a test function that also takes state and skill as input $D_\phi(s, z)$, and this test function is learned by:

$$\min_\phi \mathbb{E}_{z \sim p(Z)} \mathbb{E}_{z' \sim p(Z \neq z)} \Big[ \mathbb{E}_{s \sim p(S|z')}[D_\phi(s, z)] - \mathbb{E}_{s \sim p(S|z)}[D_\phi(s, z)] \Big]$$
$$+ \lambda \mathbb{E}_{(s,z) \sim \hat{P}(s,z)}[(\|\nabla_s D_\phi(s, z)\|_2 - 1)^2] \tag{95}$$

where the last is a Lagrange term that restricts the test function to be 1-Lipschitz with respect to $s$, and $\hat{P}(s, z)$ is the distribution of interpolated samples between states inferred by skill $z$ and states not inferred by skill $z$. Test function $D_\phi$ learned by this loss can be used to estimate the dual form Peyré & Cuturi (2019) of the 1-Wasserstein distance between states inferred or not inferred by any skill $z$. Then, pertaining skill $z$ by the intrinsic reward $\mathbb{E}_{z' \neq z}[D_\phi(s, z) - D_\phi(s, z')]$ could maximize WSEP.

We can see that both eq. (91) and eq. (95) share similarities with the discriminator loss in generative adversarial networks (GANs) (Goodfellow et al., 2020) and Wasserstein GANs (Arjovsky et al., 2017). Therefore, it is encouraging better separability between states inferred by different skills thus resulting in more distinctive skills.

### H.3 PRACTICAL IMPLEMENTATION FOR PWSEP

The key to implementing algorithm 1 for PWSEP is how to efficiently compute the projection

$$\text{PWSEP}(i) = \min_\lambda W\Big(p^{\pi_{\theta_i}}(S|z_i), \sum_{z_j \in \mathcal{Z}_i} \lambda^j p^{\pi_{\theta_j}}(S|z_j)\Big) \tag{96}$$

We can see that this problem is convex for $\lambda$. The derivative $\frac{\partial \text{PWSEP}(i)}{\partial \lambda}$ can be approximated by sensitivity analysis of linear programming $\min W\Big(p^{\pi_{\theta_i}}(S|z_i), \sum_{z_j \in \mathcal{Z}_i} \lambda^j p^{\pi_{\theta_j}}(S|z_j)\Big)$. However, with a larger batch size $B$ of $(s, z)$ samples, it could be infeasible to obtain this derivative from the implicit function theorem and KKT conditions like the approach from OptNet (Amos & Kolter, 2017), because each derivative calculation requires solving an inverse or pseudo-inverse of a $B$-dimensional matrix. We have compared it with the derivative-free method CMA-ES (Hansen & Ostermeier, 2001), finding that although CMA-ES computes the Wasserstein distance multiple times for one update of $\lambda$, the total time cost of CMA-ES is much more efficient. Besides, because we have separated parameter $\theta_i$ for each skill policy, and $p^{\pi_{\theta_j}}(S|z_j)$ of other skills remain unchanged when updating $\theta_i$, the optimal $\lambda$ after an $\theta_i$ update would be close to the optimal $\lambda$ before the update. Therefore, the computation for $\lambda$ to be optimal could be large at the beginning then it gets small.

### H.4 CHOICE OF TRANSPORTATION COST FOR WASSERSTEIN DISTANCE

Wasserstein distance provides stable and smooth measurements when the transportation cost can be defined meaningfully. For example, in environments with physical dynamics, the transportation cost between states could be defined as the physical distance, in certain discrete environments, it can be defined as the Manhattan distance. These are not only more intuitive but also provide smooth and stable measurements with meaningful information on how difficult it is for an agent to reach from one state to the other. However, the choice of them requires prior knowledge of the environment. For completely unsupervised transport cost design, our idea for future work is to learn a representation so that the basic distances like L2 in the representation space can capture how difficult it is to travel from one state to another, which could be related to previous work about graph Laplacian representation (Wu et al., 2019). Their method uses spectral graph drawing (Koren, 2003) to learn a state representation so that dynamically consecutive states have small L2 distance to each other in the representation space.

### H.5 PRACTICAL NON-PARAMETRIC ESTIMATION OF LSEPIN

LSEPIN can be approximated by particle-based entropy (Singh et al., 2003a; Liu & Abbeel, 2021b):

$$I(S; \mathbf{1}_z) \approx \hat{H}_{\text{PB}}(S) - \hat{H}_{\text{PB}}(S|\mathbf{1}_z). \tag{97}$$

Where $\hat{H}_{\text{PB}}(\cdot)$ is the particle-based entropy that can be estimated as:

$$\hat{H}_{\text{PB}}(S) := \sum_{i=1}^{n} \log \left( c + \frac{1}{k} \sum_{s_i^{(j)} \in N_k(s_i)} \|s_i - s_i^{(j)}\| \right), \tag{98}$$

where $N_k(\cdot)$ denotes the $k$ nearest neighbors around a particle, $c$ is a constant for numerical stability (commonly fixed to 1).

The second term of Equation 97 is the state entropy conditioned on knowing whether skill equals $z$, and it is defined as

$$H(S; \mathbf{1}_z) = H(S|Z = z) + H(S|Z \neq z). \tag{99}$$

The first term of Equation 99 can be estimated by sampling from the states generated with skill $z$. The second term is a little more tricky for parametric methods that try to approximate $\log p(s|z)$, but for particle-based entropy, it can still be conveniently estimated by sampling states from all skills not equal to $z$.

## I EMPIRICAL VALIDATION

We provide some empirical examples to validate the proposed theoretical results. For more intuition, we consider the skill trajectories in a visualized maze environment with continuous state and action spaces from Campos et al. (2020) and a higher dimensional Mujoco Ant environment (Todorov et al., 2012). Maze navigation and high-dimensional control are major test grounds for previous MISL methods (Eysenbach et al., 2019; Kim et al., 2021; Park et al., 2022b).

### I.1 CONTINUOUS MAZE ENVIRONMENT

To get an intuitive understanding of what kind of skills are encouraged by our proposed metrics, we first look at the trajectories in the maze environment.

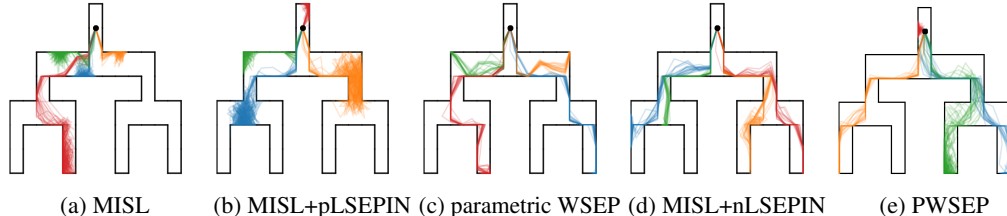

(a) MISL     (b) MISL+pLSEPIN   (c) parametric WSEP   (d) MISL+nLSEPIN     (e) PWSEP

Figure 7: Trajectory examples: Trajectories of different skills are in different colors. (a) is learned by typical MISL with state entropy for exploration (Lee et al., 2019; Liu & Abbeel, 2021b), while (b)(c)(d) are proposed in appendix H our paper. (b) is learned by MISL with state entropy and parametric LSEPIN proposed in appendix H.1. (c) is learned by parametric WSEP proposed in appendix H.2. (d) is learned by MISL with state entropy and non-parametric LSEPIN estimated by particle-based entropy (Singh et al., 2003a; Liu & Abbeel, 2021a). (e) is learned by PWSEP, the projection in eq. (15) is found by the CMA-ES algorithm (Hansen & Ostermeier, 2001) as described in appendix H.3.

The corresponding metrics for the agents in fig. 7 are listed in table 3. For stable estimation, the metrics are estimated by non-parametric approaches using a large number of samples. Metrics related to mutual information ($I(S; Z)$ and LSEPIN) are estimated by particle-based entropy eq. (98) with $n = 4000, k = 2000$. WSEP is estimated by sliced Wasserstein distance (Kolouri et al., 2019) using 4000 samples of $(s, z)$. The magnitude of particle-based entropy depends on the number of samples $n$ because it is a summation instead of a mean, while the magnitude of sliced Wasserstein distance is a mean and it is less affected by the number of samples. As mentioned in appendix G.4, the sum of PWSEP($i$) in eq. (76) can not serve as evaluation metrics like LSEPIN and WSEP, so we consider to evaluate with $I(S; Z)$, LSEPIN and WSEP.

Table 3: Metrics for the agents in fig. 7.

|  | $I(S; Z)$ | LSEPIN | WSEP |
|---|---|---|---|
| (a) MISL | 787.9 | 161.5 | 15.3 |
| (b) MISL+pLSEPIN | 1060.6 | 244.6 | 23.5 |
| (c) Parametric WSEP | 1090.0 | 286.7 | 30.9 |
| (d) MISL+nLSEPIN | 965.6 | 243.4 | 29.3 |
| (e) PWSEP | 942.3 | 193.2 | 31.6 |

Parametric LSEPIN encourages the discriminability among skills, which is shown by comparing figs. 7a and 7b. The agent learned with parametric LSEPIN has less overlapping among skills thus better discriminability. WSEP encourages more Wasserstein distances between skills and skills with more distances from each other are also more discriminable. Because particle-based entropies are calculated by the distances between samples, which is the same as the transportation cost of Wasserstein distance, the mutual information calculated with particle-based entropies measures not only discriminability but also some kind of "distance". This is why the WSEP agent has high particle-based LSEPIN. PWSEP learns the skills by the order of the Matplotlib (Hunter, 2007) color cycle, so the red skill is learned in the end, so it goes away from other skills instead of going downwards to the undiscovered branch of the tree maze, which could be a local optimum for PWSEP(i). In practice, because of local optimums and a limited number of skills, despite its favored theoretical properties, the PWSEP algorithm might perform the best.

By comparing figures in fig. 7 and table 3, we can see that LSEPIN and WSEP capture the diversity and separatability of skills.

## I.2 ANT ENVIRONMENT

Ant is one of the most common environments for URL (Sharma et al., 2020; Park et al., 2022a; Kim et al., 2021). It is challenging high-dimension continuous control but the skills can be visualized by top-down view of x-y dimension. This environment has downstream tasks that require navigating the ant agent to designated positions. We compare the top-down visualizations, downstream task performance, and our proposed metrics to see whether the correlation claimed by our theorems exists.

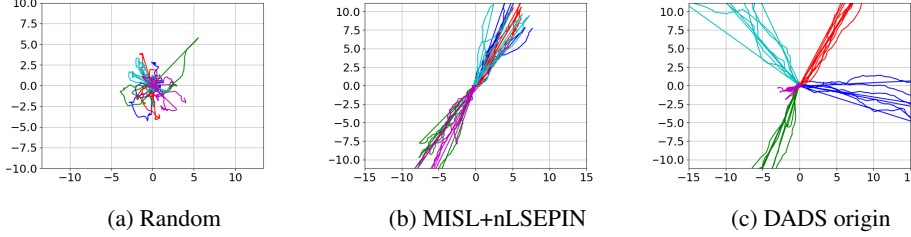

(a) Random                    (b) MISL+nLSEPIN                    (c) DADS origin

Figure 8: Trajectory examples: Trajectories of different skills are in different colors. (a) is a randomly initialized agent. (b) is pre-trained by MISL with non-parametric LSEPIN. (c) is pre-trained by the original implementation of DADS with prior knowledge and specific tuning.

Figure 8 shows the top-down view of skills of three URL agents. Their skills are in different colors. Their corresponding metrics are shown in table 4. DADS origin was implemented with prior knowledge and specific tuning. For example, there is a specific logarithmic transformation of its intrinsic rewards which does not apply to our implementations.

"R" is the downstream task performance, MSEPIN is the median of $I(S; \mathbf{1}_z)$. Similarly, WSEP is estimated by sliced Wasserstein distance. $I(S; \mathbf{1}_z)$ and $I(S; Z)$ are estimated by particle-based entropy.

Table 4: Metrics for the agents in fig. 8.

| Agent | Dimension | $I(S;Z)$ | LSEPIN | MSEPIN | WSEP | R |
|---|---|---|---|---|---|---|
| (a) Random | All | 20.8 | 6.1 | 10.2 | 4.5 | -0.95 |
| | x-y | 129.2 | 12.3 | 24.5 | 10.4 | |
| (b) MISL w nLSEPIN | All | 127.6 | 20.1 | 23.9 | 15.3 | -0.46 |
| | x-y | 288.7 | 53.6 | 60.6 | 50.4 | |
| (c) DADS | All | 248.5 | 31.0 | 71.7 | 26.2 | -0.37 |
| | x-y | 639.4 | 55.9 | 178.8 | 95.1 | |

The correlation coefficient between downstream task performance and the metrics are shown in table 5. We can see that there is a strong correlation between disentanglement metrics (LSEPIN/WSEP) and downstream task performance. This empirically validates our proposed theorems which claim a correlation between disentanglement metrics and downstream task performance.

Table 5: Correlation coefficient between downstream task performance and the metrics

| Dimension | $I(S;Z)$ | LSEPIN | MSEPIN | WSEP |
|---|---|---|---|---|
| All | 0.92 | 0.95 | 0.77 | 0.93 |
| x-y | 0.83 | 0.99 | 0.78 | 0.92 |

### I.3 SCALABILITY OF PARAMETRIC OBJECTIVES

Although we are not focusing on the empirical performance, we find that the proposed practical algorithm design in appendix H has the scalability for a potentially larger number of skills.

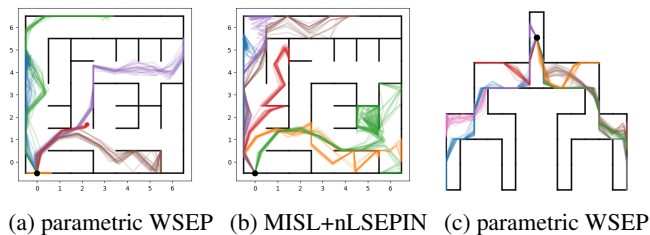

(a) parametric WSEP    (b) MISL+nLSEPIN    (c) parametric WSEP

Figure 9: Trajectory examples: Trajectories of different skills are in different colors. (a) is learned by only parametric WSEP. (b) is learned by MISL with non-parametric LSEPIN and state entropy. (c) is pre-trained by only parametric WSEP.

The agents in fig. 9 are learned by 8 skills by our introduced practical algorithms. Empirical algorithm design is not our main focus but we can see that optimizing parametric WSEP (eq. (95)) alone has comparable state coverage as MISL with state entropy for exploration

### I.4 ABOUT UNSUPERVISED REINFORCEMENT LEARNING BENCHMARK

There exists an unsupervised reinforcement learning benchmark (URLB) (Laskin et al., 2021) using DeepMind Control environments, but their downstream task setting is not suitable for evaluating vanilla MISL methods that approximate and maximize $I(S;Z)$. MISL with $I(S;Z)$ is essentially partitioning the state space and labeling each part with a skill latent $z$. The downstream task in Laskin et al. (2021) focuses on movements like "walk", "run" and "flip", which require learning a specific sequence of state progression instead of reaching a certain part of the state space. So it can be considered that the downstream task setting in URLB is too complicated for current MISL methods. And the downstream finetuning performance of pure exploration methods like ProtoRL Yarats et al.

(2021), APT Liu & Abbeel (2021a), RND Burda et al. (2019) exceeds MISL methods from Lee et al. (2019); Eysenbach et al. (2019); Liu & Abbeel (2021b) for by large margin, this might be because the state distributions of the MISL learned skills cover fewer states that are useful for the downstream task than the pure exploration methods. The CIC method Laskin et al. (2022) optimizes a modified MISL objective $I(\tau; Z)$ and outperforms the pure exploration methods, they claim CIC promotes better diversity and discriminability than previous MISL methods, which also accords with our results.

