# OpenReview forum: "Task Adaptation from Skills: Information Geometry, Disentanglement, and New Objectives for Unsupervised Reinforcement Learning"
_ICLR.cc/2024/Conference — ICLR 2024 spotlight_

### Official Review · Reviewer_LRDL · 2023-10-20

**Soundness:** 4 excellent
**Presentation:** 3 good
**Contribution:** 4 excellent
**Rating:** 8
**Confidence:** 5

**Summary:**

This work targets on the Information Geometry in unsupervised Reinforcement Learning, especially Mutual Information Skill Learning (MISL). On the basis of previous work, this work first considers the diversity and separability of learned skills. As MISL can not guarantee these properties, this work proposes LSEPIN to measure the disentanglement, and then shows the connection between LSEPIN and downstream task adaptation cost. Moreover, this work investigates the information geometry of Wasserstein distance based skill learning methods. Finally, this work proposes PWSEP and theoretically shows that it can discover all optimal initial policies.

**Strengths:**

- Definitely an important problem to be tackling! Previous work[1] has established the connection between skill-based unsupervised RL and information geometry. [1] has shown that KL-based metric can only find skills with the ''largest radius'' and how to find all vertices of the skill polytope is still an open challenge. This work has elegantly solved this challenge and I believe it will be of interest to researchers of Unsupervised RL.

- The description of the article is very clear, and the introduction to related work is very specific.

- I have basically read all the proofs of the theorems, which are well-written.

Thanks to the authors for putting in the effort in doing this work!

[1] Eysenbach, Benjamin, Ruslan Salakhutdinov, and Sergey Levine. "The information geometry of unsupervised reinforcement learning." arXiv preprint arXiv:2110.02719 (2021).

**Weaknesses:**

Overall, I think this paper is well written and its contribution is solid. I do not find clear weaknesses but I still have some questions (see Questions). I will adjust my score accordingly based on the author's response and other reviewers' comments.

**Questions:**

- This paper claims that "it is possible for WSEP to discover all vertices of the feasible polytope". As the theoretical results can only show that all learned skills satisfy $p(z) > 0$ rather than WSEP can learn all $p(z) > 0$ skills, there is no direct evidence to suggest that WSEP can indeed find more skills than MISL (i.e., skills without maximum “distances”). Can authors show that WSEP can find more skills or even all skills? I think empirical results or theoretical results even in simple cases will be really helpful.

- All experiments are provided in the Appendix. It seems that providing the main experiments and analyses in the main text can help the readers to better understand this work.

- What will happen if we change the W-distance to other distance metrics in PWSEP(i)? In my opinion, it seems that the proof of Thm 3.5 holds for any distance metric (owns Non-negativity, Symmetry, and Triangular Inequalities). Is it right? Or are there some special properties of W-distance necessary for proving Thm 3.5?

- It's better to provide some proof sketch of theorems in the main text, like Thm 3.5.

---

> ### Author Response · Authors · 2023-11-15
> **First response**
>
> We sincerely thank you for your thoughtful review and recognition of our work. We are delighted to address your questions. Please see our response below:
>
> ## **Q1:** About whether "WSEP can find more skills or even all skills"
>
> **A1:** We consider that, by 'skills' you are referring to those at the vertices.
>
> By *"it is possible for WSEP to discover all vertices of the feasible polytope"* we mean that WSEP is not restricted by maximum "distances", so it might discover the vertices that can not be discovered by MISL in certain cases.  For example, in the situation shown by the figure from [this anonymous link](https://anonymous.4open.science/r/iclrrebfig-4B58), MISL can only learn $z_1,z_2$. Even when the skill number is set to be $|Z|=3$, the third skill learned by MISL will be duplicated at $z_1$ or $z_2$ but not lie at $z_3$. Instead, WSEP can discover all 3 vertices in this case.
>
> However, as mentioned in Section 3.3.4 and appendix F.6, unlike PWSEP, optimizing WSEP is not guaranteed to **always** discover all vertices. Among the vertices that can not be discovered by WSEP, there can also be ones with maximum "distance" that can be discovered by MISL. This motivates us to look into PWSEP, which is guaranteed to always discover all vertices with enough skills.
>
>
> ## Q3: What will happen if we change the W-distance to other distance metrics in PWSEP(i)?
>
> Thm 3.5 holds for any distance metric that owns "Non-negativity, Symmetry, and Triangular Inequalities" and strict convexity.
>
> We briefly discussed this topic in appendix F.4 after the proof of theorem 3.5. Although total variation distance and Hellinger distance are also true distance metrics, they're less used in RL compared to KL divergence and Wasserstein distance, resulting in limited research on their efficient approximations for state distributions. In addition, we haven't found valid literature to support the strict convexity of total variation distance and Heilinger distance. Additionally, Wasserstein distance has the potential to take advantage of the choice of transportation cost to provide smooth and stable measurements with meaningful information as discussed in appendix G.4
>
> ## Q2&Q4: About experimental results in the main text and proof sketch
>
> Thank you for your valuable suggestion on enhancing our paper's presentation. We will certainly take your feedback into account.
>
> ---
> Thanks again for your time and attention!

---

> > ### Comment · Reviewer_LRDL · 2023-11-16
> > **Thanks for your reply**
> >
> > I have read your reply and my Q1 still holds. I fully understand that MISL can only find skills restricted by maximum "distances" where WSEP **might** find other skills. My question is: **Is there a situation in which WSEP indeed finds more skills than MISL?** More specifically, can we find an MDP, maybe very simple, where WSEP can find more skills than MISL? The proposed figure in the rebuttal is not a strict example, right?

---

> ### Author Response · Authors · 2023-11-16
> **Second response**
>
> Thanks for the fast reply!
>
> We hope the [figure](https://anonymous.4open.science/r/iclrrebfig-4B58) provides the intuition that WSEP can learn the vertices that are not with maximum "distances".
>
> Here is a simple MDP that produces the feasible state distribution polytope in the [figure](https://anonymous.4open.science/r/iclrrebfig-4B58), and the quantitative results below shows why in this case, WSEP learns more vertices than MISL.
>
> ---
> ## An MDP example that produces the polytope in the figure
>
> An MDP with $|\mathcal{S}|=3$ for states and $|\mathcal{A}|=3$ for actions.
> The transition matrix for action $a_1$ is:
>
> $\begin{pmatrix}
> 0.2 & 0.7 & 0.1\\\\
> 0.2 & 0.7 & 0.1\\\\
> 0.2 & 0.7 & 0.1
> \end{pmatrix}$
>
> The transition matrix for action $a_2$ is:
>
> $\begin{pmatrix}
> 0.2& 0.1& 0.7\\\\
> 0.2& 0.1& 0.7 \\\\
> 0.2& 0.1& 0.7
> \end{pmatrix}$
>
> The transition matrix for action $a_3$ is:
>
> $\begin{pmatrix}
> 0.4 & 0.3 & 0.3\\\\
> 0.4 & 0.3 & 0.3 \\\\
> 0.4 & 0.3 & 0.3
> \end{pmatrix}$
>
> Because the transition probabilities only depend on actions, state distribution $p(S)$ is determined by distribution $p(A)$:
> $p(S) = p(a_1)[0.2,0.7,0.1]+p(a_2)[0.2,0.1,0.7]+p(a_3)[0.4,0.3,0.3]$
>
> We can see $p(A)$, in this case, is the convex coefficient of three probabilities, so the feasible state distribution is in the convex polytope $\mathcal{C}$ with these three probabilities being the vertices of $\mathcal{C}$.
>
> This MDP accords with the figure, where we have labeled the vertices with $z_1,z_2,z_3$,
>
> $p(S|z_1)= [0.2, 0.7, 0.1]$
>
> $p(S|z_2)= [0.2, 0.1, 0.7]$
>
> $p(S|z_1)= [0.4, 0.3, 0.3]$
>
>
>
> ## Quantitative comparision of I(S;Z) and WSEP
>
> >### MISL only learns skills at $z_1,z_2$:
>
> For this MDP, the unique center of the "circle" with "maximum radius" (We showed that it's uniquely determined by the MDP in appendix D) would be $[0.2,0.4,0.4]$.
>
> MISL would learn only $z_1,z_2$ with $p(z_1)=p(z_2)=0.5,p(z_3)=0$ to have the average state distribution $p(S)=\sum_z(S|z)$ at the unique center $[0.2,0.4,0.4]$.
>
> By this solution, $I(S;Z)$ is maximized by $I(S;Z)=\sum_z p(z)D_{KL}(p(S|z)\|p(S))\approx 0.253$.
>
> Putting weight on the other vertex $z_3$ would lower $I(S; Z)$. For example, if $p(z_1)=p(z_2)=0.45,p(z_3)=0.1$, the average $p(S)$ would be $[0.22, 0.39, 0.39]$ which is not the center of the maximum "circle" any more, and their $I(S; Z)\approx 0.237$.
>
> >### WSEP can learns skills at all vertices $z_1,z_2,z_3$:
>
> For WSEP, when the skill number is set to $|Z|=3$, when two skills are at $z_1,z_2$ like MISL, putting the last skill at $z_1$ or $z_2$ would have lower WSEP than putting the last skill at $z_3$, due to the triangle inequality.
>
> Suppose the transportation cost between every two states is 1, then
>
> $W(p(S|z_1),p(S|z_2))=0.6$
>
> $W(p(S|z_1),p(S|z_3))=W(p(S|z_2),p(S|z_3))=0.4$
>
> WSEP for $2$ skills at $z_1$ and $1$ skill at $z_2$ would be $2.4$,
>
> WSEP for $2$ skills at $z_2$ and $1$ skill at $z_1$ would also be $2.4$.
>
> WSEP for $3$ skills at $z_1,z_2,z_3$ respectively would be $2.8$, and this is the solution for maximizing WSEP for this case (By convexity, moving any of the three vertices would lower its distance to the others).
>
> Therefore, WSEP would favor learning all $3$ vertices instead of only $2$ skills at $z_1,z_2$ like MISL. In this case, MISL can not discover $z_3$ that is not with maximum "distances" but WSEP can.
>
> ---
> Thanks again and we hope this addressed the question!

---

> > ### Comment · Reviewer_LRDL · 2023-11-17
> > **Thanks for your reply**
> >
> > I have roughly checked the example and believe it is reasonable. Also, I believe that adding a more detailed version of this example (for example, it is better to strictly show that [0.2, 0.4, 0.4] is exactly the center of the "circle") to the paper will make the paper more solid. I would like to keep my score and believe that this work will be a valuable contribution to the community.

---

### Official Review · Reviewer_j7aV · 2023-10-30

**Soundness:** 2 fair
**Presentation:** 1 poor
**Contribution:** 3 good
**Rating:** 6
**Confidence:** 3

**Summary:**

This paper studies the geometry of state distributions learned with mutual information skill learning for the purpose of theoretical task adaptation analysis. The authors propose Least SEParability and INformativeness (LSEPIN) to measure the diversity and separability and show a relationship with worst-case adaptation cost (WAC). The authors theoretically prove the relationship between the optimization of LSEPIN and Similarly, the authors also propose Wasserstein based distance metric WSEP which is more suitable in symmetric polytope to measure the separability of skill. Similar to LSEPIN and WAC, the relationship between WSEP and Mean Adaptation Cost is studied.

**Strengths:**

* The geometry perspective of task adaptation is interesting.
* Studies on the geometry is promising and can motivate readers.
* The theoretical results are well justified.

**Weaknesses:**

* Some Definitions of words are not defined well (diversity and separability).
* The flow of the paper is weakly organized and hurts readability (the first topic is LSEPIN and the second WSEP.) The paper have no discuss on other perspectives.
* The findings from the theoretical derivation are not surprising. WAC and adaptation cost.
* Most contents have high dependency with appendix. Although the authors studied various components, they are not well organized.

**Questions:**

* Here are questions that we want to discuss with the authors.
* We might incorrectly understand the contribution of this work. Why the theoretical derivation of Theorem 3.1 is important for a task adaptation?
    * What can we additionally learn from the theorem 3.1 or how can we use the theorem for other task adaptation works? For my understanding, the relationship: increasing LSEPIN results in lower WAC.
    * Why measuring $\min_z I(S;z)$ can be used to measure diversity and separability? I guess two features should be computed among skills, but the $\min_z I(S;z)$ is just a mutual information for a single code.
    * In LSEPIN, the metric measures the least skill code. I guess an abundant skill code may hurt the calculation. Isn't it problematic? How could you ensure that all the codes are meaningful in the computation of LSEPIN?
    * WAC is measured with the worst state distribution. how this could be meaningful and practical? That is, why we need to measure the worst case adaptation? To the best of my knowledge,  skill adaptation is applied to the state distributions which are similar to the state distribution for the target skill. Therefore, the importance of $\max_p$ part in WAC is not persuasive.
    * Could you please additionally describe the necessity of symmetry and triangle inequality?
    * What is the limitation of WSEP?
    * Could this study can be combined with in-distribution and out-distribution perspective of task adaptations?

[Overall]
Although the authors conducted several theoretical derivations and analysis, the choice of metrics and the relationship between them have little meaning. Mostly because the task adaptation is might assume close state distribution $p$ for a skill $p(s|z)$. However, the authors study the worst-case adaptation. Additionally, the organization of the paper is not well constructed and hard to follow.*

---

> ### Author Response · Authors · 2023-11-15
> **First response (1/2)**
>
> We thank the reviewer for your valuable feedback and your willingness to engage in discussion. We would like to address your concerns and answer your questions. Please see the following for our response.
>
> ---
> ## 1. Regarding the concerns related to the "WAC" cost and Theorem 3.1
> About:
> >- *[Overall] Although the authors conducted several .. the authors study the worst-case adaptation.*
> >- *WAC is measured with the worst  ... the importance of $max_p$ part in WAC is not persuasive.*
> >- *What can we .. For my understanding, the relationship: increasing LSEPIN results in lower WAC.*
> >- The findings from the theoretical derivation are not surprising. WAC and adaptation cost.*
>
> ### 1.1 Clarification of the definition of WAC
> To address these concerns, we would like to first clarify the definition of the WAC cost and mitigate potential misunderstandings. The WAC is defined as:
> $$\text{WAC} = \max_r \min_{z\in \mathcal{Z}^*}D_{KL}({p(S|z)}\|{p^r})$$
> $p^r$ is the optimal state distribution for downstream task $r$, and $\mathcal{Z}^*$ is the set of learned skills.
>
> WAC is indeed considering adaptation from **closest** skills in the set of learned skills because of $\min_{z\in \mathcal{Z}^*}D_{KL}({p(S|z)}\|{p^r})$. The $\max_r$ in WAC means to choose a downstream task $r$ such that its optimal state distribution $p^r$ is the far from all learned skills, even for the skill $z^*=argmin_{z\in \mathcal{Z}^*}D_{KL}({p(S|z)}\|{p^r})$ that is **closest** to $p^r$.
>
>
> ### 1.2 Practicality and meaningness
> In practice, although we can adapt from the closest skill after knowing the downstream task $r$, we do not have prior knowledge about $r$ during unsupervised pre-training. Therefore, we can only prepare for the worst downstream task (the one far from even the closest learned skill).
>
> Theorem 3.1 and the theoretical results in Section 3.2 shed light on what kind of skills are favored for downstream task adaptation and how to quantitatively measure them.
>
> ### 1.3 About the contribution of our findings
>
> Following the above clarification of the WAC definition, we hope it is clear now that our analysis is dealing with the fundamental question for URL: "How the learned skills can be used for downstream task adaptation, and what properties of the learned skills are desired for better downstream task adaption?"
>
> > The findings ... not surprising.
>
> The findings may appear "not surprising", given that promoting diversity and separability of learned skills has been an intuitive heuristic in prior practical algorithms [2][3]. However, our unique contribution lies in offering a theoretical justification for this heuristic.
>
> ---
> ## 2. Regarding concerns related to "diversity and separability"
> About:
> >- *Why measuring $min_z I(S;z)$ can be used to measure diversity and separability? ..., but the $min_z I(S;z)$ is just a mutual information for a single code.*
> >- *Some Definitions of words are not defined well (diversity and separability).*
>
>
> First of all, LSEPIN is defined as $\min_z I(S;\mathbf{1}_{z})$, its mean is not $I(S,Z)$ (details in "formal difference" of appendix B.2).
>
> As we have mentioned at the beginning of Section 3:
> *"Separability means the discriminability between states inferred by different skills."*
> In inequality (61) of appendix E, we show that increasing $I(S;\mathbf{1}_{z})$ promotes the KL divergence between $p(S|z)$ and $p(S|Z\neq z)$(average state distribution of non-$z$ skills). Also discussed in appendix E, higher KL divergence between $p(S|z)$ and $p(S|Z\neq z)$ means skill $z$ is more distinctive and has less overlap with other skills, so this means better separability.
>
> For a set of skills, they are diverse because all skills in the set are distinctive and have little overlap with each other (they are "far" from each other in terms of KL divergence). The standard MISL objective $I(S;Z)$ can not guarantee the seperability of each skill, as discussed in Section 3 after the list of informal results, so MISL without LSEPIN can not guarantee diversity.
>
> ---
> ## 3. Regarding why focus on single skill code
> >In LSEPIN, the metric measures the least skill code. ... How could you ensure that all the codes are meaningful in the computation of LSEPIN?
>
> It is important to ensure every learned skill is distinctive and separable from others. High $I(S;\mathbf{1}_{z})$ means that this skill covers a specific region of the state space that is less covered by other skills. If it is not separable from other skills, its state coverage could have a huge overlap with other states, so deleting this skill would make no difference for exploration or downstream task adaptation.
>
>
> In [1], they implement many existing algorithms with a low number of skills, eg. only 4 skills for SMM. If one of its skills exhibits a low $I(S;\mathbf{1}_{z})$, it may have a huge overlap with other skills, leading to a situation where 25% of the skills are underutilized, offering no meaningful contribution to exploration and downstream task adaptation.

---

> > ### Author Response · Authors · 2023-11-15
> > **First response (2/2)**
> >
> > ## 4. Regarding the necessity of symmetry and triangle inequality
> > > Could you please additionally describe the necessity of symmetry and triangle inequality?
> >
> > This topic is discussed in Appendix B.3. We showed an example of what could happen if we replace the Wasserstein distance with KL divergence in appendix F.7. The example in appendix F.7 shows that if we replace the Wasserstein distance in WSEP with KL divergence, due to the lack of triangle inequality, maximizing this metric could result in skills too close together, harming diversity.
> >
> > Symmetry is important when comparing distances between different pairs of points, for example, we have pair $a,b$ and pair $b,c$ and a non-symmetric measure $d$. Without symmetry, $d(a,b)\neq d(b,a)$ and $d(c,d)\neq d(d,c)$. Then it's possible that there exists a situation where $d(a,b)>d(c,d)$ but $d(b,a)<d(d,c)$. From the $d$ measurement you cannot know whether pair $a,b$ is more separable than pair $c,d$.
> >
> > Besides, state distributions of two different skills could share different domains, which is a problem for a well-defined (not infinity) KL divergence.
> >
> > ---
> > ## 5. Regarding limitations of WSEP
> > > What is the limitation of WSEP?
> >
> > - Theoretically, as discussed in Section 3.3.4 and appendix F.6, although WSEP also discovers vertices, it is not guaranteed to discover all $|\mathcal{V}|$ vertices with only $|\mathcal{V}|$ skills. This motivates us to propose the PWSEP algorithm to solve the vertex discovery problem using only $|\mathcal{V}|$ skills.
> >
> > - Practically, the implementation of Wasserstein distance depends on the choice of transportation cost. For continuous control, it is common that the transportation cost is chosen as the L2 norm. However, the L2 norm between two states might not accurately measure how difficult it is to travel from one state to the other, because there might be obstacles between them.
> >
> > As discussed in appendix G.4, an idea for future work is to learn state representations so that the L2 norm in the representation space can reflect the actual difficulty of traveling from one state to the other.
> >
> > ---
> > ## 6. Regarding the organization of the paper and readability
> > >- The flow of the paper is weakly organized and hurts readability (the first topic is LSEPIN and the second WSEP.) The paper have no discuss on other perspectives.
> > >- Most contents have high dependency with appendix. Although the authors studied various components, they are not well organized.
> >
> > The logic flow of our paper is like this:
> > 1. We try to answer the fundamental question of URL: "How the learned skills can be used for downstream task adaptation and what properties of the learned skills are desired for better downstream task adaption?”
> > 2. We found that LSEPIN captures the desired properties necessary for preparing skills for downstream task adaptation.
> > 3. We found that LSEPIN and MISL are essentially optimizing KL divergences and MISL with LSEPIN also has limitations like the one mentioned in remark 3.2.1, this inspires us to investigate whether we can overcome the limitations of MISL by optimizing true metrics between state distributions.
> > 4. We found that WSEP and PWSEP with true metric Wasserstein distance can overcome the limitation and can discover potentially optimal skills that can not be discovered by MISL.
> >
> > Thank you for pointing out the readability and organization issues that can be further enhanced. We could replace some sentences with tables or diagrams.
> >
> > ---
> > ## 7. Regarding the relation to in-distribution and out-distribution perspective of task adaptations
> > >Could this study can be combined with in-distribution and out-distribution perspective of task adaptations?
> >
> > First of all, in unsupervised RL setting, the target task distribution is unknown during training, it can only learn by its intrinsic motivations like intrinsic rewards for maximizing $I(S; Z)$ or WSEP. Therefore, it is unlikely that the learned skills cover the target task distribution. The URL setting should be closer to the out-distribution meta RL setting, where the skills learned by intrinsic rewards can be considered as the training tasks and the downstream tasks can be considered as out-of-distribution test tasks. One future research idea could be how to combine the URL approaches with training tasks so that the pretrained agent can be well-prepared for out-of-distribution downstream tasks
> >
> >
> >
> > ## Reference
> >
> > [1] laskin 2021， URLB: Unsupervised Reinforcement Learning Benchmark
> >
> > [2] Eysenbach 2019, Diversity is all you need: Learning skills without a reward function.
> >
> > [3] laskin 2022，CIC: Contrastive Intrinsic Control for Unsupervised Skill Discovery
> >
> > ---
> > Thanks again for your constructive feedback!

---

> ### Author Response · Authors · 2023-11-19
>
> Thanks again for your valuable feedback on our paper. Our rebuttal addresses the concerns, especially the ones related to the WAC cost definition. By WAC, we actually consider the practical adaptation procedure you mentioned, which is to adapt from the 'closest' skill. We hope this clarifies our main contribution.
>
> We wonder whether you have any remaining concerns, we are looking forward to addressing any additional concerns you may have.

---

> > ### Comment · Reviewer_j7aV · 2023-11-21
> >
> > Thank you for the detailed explanations on specific questions.
> > 1. Regarding the concerns related to the "WAC" cost and Theorem 3.1
> > Thank you for clarifying the definition. The definition based on the unknown optimal distribution is persuasive.
> > 2. Thank you for clarifying the definition of LSEPIN and separability. Please refer B.2 in the paper. The definition of $I(S,1_z)$ helps me understanding the definition of separability.
> > 3. Thank you additional explanation on the meaning of separable skill learning.
> > 4. I understand the impact of the lack of triangle inequality. Thank you for the explanation.
> > I appreciate your effort to provide additional comments on the questions, especially in sections 5.6 and 5.7. Thank you
> > ----
> > Here are additional comments to clarify the problem tackled in this work.
> > Two terms **"diversity" and "separability"** are used jointly in this work. In my understanding diversity is about $I(S;Z)$ [1] which measures the coverage of skills, while the separability is defined with $D_{KL}(P(S|Z=z)||P(S|Z\ne z))$.
> > To the best of my understanding, the main contribution is on separability. Is the contribution of this work is on both properties?
> > (related to question 6, organization of the paper)
> > I also checked the reviews from other reviewers.  I also agree that the inclusion of the main experimental results to improve readability.
> > [1] Eysenbach, Benjamin, et al. "Diversity is All You Need: Learning Skills without a Reward Function." International Conference on Learning Representations. 2018.

---

> > > ### Author Response · Authors · 2023-11-21
> > > **Second response**
> > >
> > > Thanks for the reply and your valuable perspective.
> > > We consider separability to be the prerequisite of diversity. Without separability, even with a large number of skills, they could be close to each other and only cover a small region of the feasible state distribution polytope, so skills need to be distinctive and separable first before they can be diverse.
> > >
> > > We showed at the beginning of Section 3 with the example of Fig. 2, that maximizing $I(S; Z)$ alone does not necessarily guarantee separability consequently resulting in limited diversity.
> > >
> > > The approach from [1] was based on the heuristic: Maximizing $I(S; Z)$ and entropy of $p(Z)$ will result in diverse skills. However, they did not provide rigorous justification for this heuristic, and we showed in the response to reviewer ZXDZ with the example from [anonymous link](https://anonymous.4open.science/r/iclrreb2024-62C5/reb_fig.png) that maximizing $I(S; Z)$ even along with entropy of $p(Z)$ could result in duplicated or inseparable skills which leading to limited diversity.
> > >
> > > We genuinely value the suggestion of moving experimental results to the main paper for better readability. Since our results are mainly theoretical, there would be a trade-off between theoretical and experimental results. Nonetheless, we will carefully consider how to integrate some experiments without compromising the theoretical emphasis.

---

> ### Comment · Reviewer_j7aV · 2023-11-22
>
> Dear Authors,
>
> Thank you for the kind response.
>
> Although the paper still exhibits weaknesses in its experimental support and the organization of the flow, the responses provided have clarified the major concerns. Therefore, I would raise my score to 'weak accept (6).
>
> Sincerely,
>
> Reviewer j7aV

---

### Official Review · Reviewer_ZXDZ · 2023-11-01

**Soundness:** 3 good
**Presentation:** 2 fair
**Contribution:** 3 good
**Rating:** 8
**Confidence:** 3

**Summary:**

The paper provides a theoretical analysis of learning skills using unsupervised reinforcement learning (URL), which serves as an initialization for learning a policy for a downstream task. The paper shows that Mutual Information Skill Learning (MISL) does not guarantee diversity and separability of learned skills, and proposes to replace the uniform distribution of skills objective with the Least SEParability and INformativeness (LSEPIN) metric to promote informativeness and separability. Moreover, the paper proposes to replace the KL divergence in MISL with Wasserstein distance that exploits better geometric properties. Finally, it proposes another Wasserstein distance-based algorithm (PWSEP) that can theoretically discover all optimal initial policies.

The authors show theoretically that LSEPIN bounds the Worst-case Adaptation Cost (WAC) and show that the Wasserstein distance has better geometrical properties (such as symmetry and triangle inequality) that leads to better skill separation. In addition, the authors provide experiments to validate the proposed theoretical results in the appendix.

**Strengths:**

The paper investigates an important topic and provides a rigorous analysis of the proposed ideas. In addition, it proposes a practical algorithm that was tested empirically and demonstrates superior results compared to existing MISL methods.

**Weaknesses:**

The paper is hard to read and follow, with lots of details.

Since most of the contribution of the paper is placed in the appendix, including all experimental results, it is hard to understand and assess its contribution without reading carefully the appendix.

**Questions:**

I would like to ask the following questions:

1. Is it possible to rigorously prove that adding a loss that promotes uniform p(Z_input) to objective (1) does not promote any diversity? Or promotes less diversity than the LSEPIN loss in all cases?

2. Is there a measure for adaptation other than Worst-case Adaptation Cost?

3. Is there any advantage to the KL divergence over the Wasserstein distance? theoretically or computationally?

4. What are the limitations of the PWSEP algorithms (specifically the WSEP objective)?

---

> ### Author Response · Authors · 2023-11-15
> **First response (1/2)**
>
> We appreciate your time and attention. We thank you for the review and comments and please see our response to your questions below.
>
>
> ## **Q1:** Is it possible to rigorously prove that adding a loss that promotes uniform $p(Z_{input})$ to objective (1) does not promote any diversity? Or promotes less diversity than the LSEPIN loss in all cases?
>
> **A1:** It might be hard to derive a quantitative bound to show adding a loss like $H(Z_{input})$ to objective (1) promotes less than LSEPIN in all cases, because the diversity of state distributions $p(S|z)$ depend on not only $z_{input}$ but also parameter $\theta$ and the function class for policy $\pi_\theta$. However, we can illustrate why such loss is not guaranteed to promote diversity like LSEPIN by the following example where maximizing $I(S; Z)$ with uniform $p(Z_{input})$ results in limited diversity:
>
> For a case of with $|S|=3$, the feasible polytope $\mathcal{C}$ allows a maximum "circle" centered at $[0.2,0.4,0.4]$ with a maximum "radius" of $0.253$ as this fig in [the anonymous link](https://anonymous.4open.science/r/iclrreb2024-62C5/) shows. There are 5 vertices $v_{1:5}$ of $\mathcal{C}$, of which $v_1, v_2, v_4, v_5$ lie on the maximum "circle". Vertex $v_1$ and vertex $v_4$ are $[0.2,0.7,0.1]$ and $[0.2,0.1,0.7]$.
>
> Because $\theta$ is shared by all skills and $z=\\{\theta,Z_{input}\\}$, we can assume $p(Z)=p(Z_{input})$
>
> When there are 4 skills to learn, uniform $p(Z_{input})$ should have $p(z_{input})=p(z)=0.25$ for all $z$. In order to achieve both uniform $p(Z)$ and maximization of $I(S; Z)$, the optimal skill set $\\{z_1,z_2,z_3,z_4\\}$ should be the one containing two skills $z_1,z_2$ at $v_1$ and the other two skills $z_3,z_4$ at $v_4$, as shown in the [figure](https://anonymous.4open.science/r/iclrreb2024-62C5/reb_fig.png). Only with this skill set, the uniform average of skills $p(S)=\sum_z p(z)p(S|z)$ with $p(z)=0.25$ could be the center of the maximum "circle" $[0.2,0.4,0.4]$.
>
> We can see from this example that although skill set $\\{z_1,z_2,z_3,z_4\\}$ maximizes both $I(S; Z)$ and $H(Z_{input})$, there are two pairs of skills being not separable thus resulting in limited diversity.
>
>
> In appendix E, we have shown higher $I(S;1_z)$ explicitly increases $D_{KL}(p(S|Z\neq z) \| p(S|z))$ thus promoting $p(S|z)$ to be separable from $p(S|Z\neq z)$ (average state distribution of skills other than $z$). Therefore, LSEPIN promotes diverse skills explicitly. Compared to skills only at $v_1$ and $v_4$, MISL with LSEPIN would favor skills at $v_1, v_2, v_4, v_5$ respectively, and for $p(S)$ to be the center of maximum "circle" $[0.2,0.4,0.4]$, distribution $p(Z)$ is not necessarily unform.
>
>
>
> ## **Q2:** Is there a measure for adaptation other than Worst-case Adaptation Cost?
>
> **A2:** This question is inspiring and we have been also considering this question recently. One idea is to consider adapting from a convex combination of learned skills instead of from one of the learned skills because a good convex combination of learned skills can be "closer" to the optimal state distribution and practically feasible to obtain.
>
> For example in Figure 2 of our main paper, a convex combination of $p(S|z_1),p(S| z_5)$ could be closer to $p^r$ than $p(S|z_1)$. In practice, the convex combined state distribution $p_\lambda(S)=\sum_z \lambda_z p(S|z)$ can be obtained by sampling $z$ from the distribution $p_\lambda(z)=\frac{\lambda_z}{\sum_{z'}\lambda_{z'}}$.
>
> For the practical adaptation procedure, we can first find the $\lambda$ resulting in a $p_\lambda(S)$ with the best accumulated reward. Because we can not directly update the parameters for $p_\lambda(S)$, we could use a new parametric model to perform offline RL with data collected by $p_\lambda(S)$ and relabeled by downstream task reward function $r$. Through this approach, we concurrently distill knowledge from pretraining and execute adaptation for the downstream task.
>
> Theoretical analysis of this adaptation procedure can be an idea for future works.
>
> ## **Q3:** Is there any advantage to the KL divergence over the Wasserstein distance? theoretically or computationally?
>
> **A3:** Computationally, for non-parametric estimation with collected data, KL divergence may be preferred over Wasserstein distance. This is attributed to the fact that KL divergence can be directly estimated using samples, whereas the estimation of Wasserstein distance requires solving a linear program, introducing additional computational complexity. Theoretically, KL divergence is always strictly convex, while the strict convexity of Wasserstein distance depends on the choice of transportation cost as mentioned in our proofs in appendix F.

---

> ### Author Response · Authors · 2023-11-15
> **First response (2/2)**
>
> ## **Q4:** What are the limitations of the PWSEP algorithms (specifically the WSEP objective)?
>
> **A4:** As mentioned in Section 3.3.4, Although lemma 3.3 shows that maximizing WSEP also discovers vertices, the discovered vertices could be duplicated (shown in appendix F.6), so it might not be able to discover all $|V|$ vertices with only $|V|$ skills. This motivates us to propose PWSEP algorithm that iteratively optimizes a projected Wasserstein distance to make sure every new iteration learns a new skill.
>
> Another limitation of WSEP is discussed in appendix F.5, showing that although a higher WSEP lowers an upper bound of the adaptation cost as our theoretical analysis shows, because of the gap between the upper bound and actual adaptation cost, sometimes high WSEP could not result in lower adaptation cost.
>
> As for PWSEP, despite its favorable theoretical property of discovering all vertices, in practice, each iteration could only learn a locally optimal skill, as mentioned in the empirical results of appendix H.
>
> ---
> Thanks again for the constructive feedback!

---

> > ### Comment · Reviewer_ZXDZ · 2023-11-21
> >
> > Thank you for your dedication in answering my questions.
> >
> > 1. I appreciate your effort to provide this illustrative example. It clarifies the difference between promoting uniform distribution $p(Z_{input})$ of skills to promoting diversity.
> > In my opinion, including this example in the paper/appendix would be valuable for the reader to clarify this point.
> >
> > 2. The idea of performing adaptation from a convex combination of learned skills instead of an adaptation from one of the learned skills sounds promising and has the potential to work better in practice. A theoretical analysis of this idea will most probably lead to the same conclusion as with adapting from one of the learned skills.
> >
> > 3. If I understand correctly, since the KL divergence is always strictly convex, for complex RL environments the KL divergence could be more computationally practical - whereas the Wasserstein distance will provide an optimal solution, but with additional computational effort. Am I correct?
> >
> > 4. Could you elaborate on the impact of PWSEP learning only a **locally** optimal skill in each iteration? In your answer, please relate to the empirical results (section H) and in general.
> >
> > I have another small question about a detail that I probably missed while reading the paper - Why is your approach free from the “non-concyclic” assumption, while the previous work of Eysenbach et al. (2022) takes this assumption into account?
> > (The assumption that limits the number of vertices on the same “circle” to be |S|)
> >
> > In general, I think that the paper provides a worthy contribution to the community and should be accepted. The authors answered most of my concerns and I’m willing to increase my score.
> >
> > That being said, the readability of the paper can be improved. I think that the paper would benefit from a rigorous definition of separability and diversity of skills at the beginning, accompanied by a few sentences dedicated to motivation and examples (see point #1).
> > In addition, including the important empirical results in the main paper will motivate the applied RL community to make use of and build upon the proposed algorithms.
> >
> > Thanks again for your detailed answers.

---

> > > ### Author Response · Authors · 2023-11-21
> > > **Second response**
> > >
> > > Thanks very much for your valuable feedback and suggestions to further improve our work.
> > > 1. It's a good idea to mention this example in the main paper and include its detail in the appendix, we will definitely consider adding it in the revision.
> > > 2. Yes, it should be related to how much of the feasible polytope is covered by the convex hull of learned skills. Therefore intuitively, the learned skills should also be separable and diverse.
> > > 3. Yes, it is correct.
> > > 4. We can look at the Figure 5 (e) in the empirical result of appendix H. The red skill is learned in the end, so it goes away from all other skills instead of going downwards to the undiscovered branch of the tree maze, and it ends up in a possibly local optimum.
> > >
> > > About the question regarding the "non-concyclic" assumption:
> > > >The “non-concyclic” assumption basically restricts the solution set of MISL to be unique. Under this assumption, there is only one unique set of skills with $p(z)>0$ to maximize $I(S; Z)$. Without this assumption, there could be different sets of skills learned by MISL, and the WAC cost favors the one with the best LSEPIN.
> > >
> > > >In practice, it is common that the "non-concyclic" does not hold and the solution set for MISL is non-unique. For example, different sets of skills can all be considered to have maximized $I(S; Z)=E_z(p(S|z)||p(S))]$, as long as there is no overlapping between any two of their state distributions because KL divergence can be considered maximized when there is least overlapping between distributions. Moreover, even if the MDP satisfies the "non-concyclic" assumption, the practical solutions could be suboptimal and on a "circle" with a smaller "radius" so all points on this "circle" are within the feasible polytope, resulting in non-unique suboptimal solution sets. LSEPIN could also benefit WAC in these suboptimal cases, as discussed in appendix C.4.
> > >
> > > ---
> > >
> > > Regarding the presentation, we greatly appreciate the suggestion to add "a rigorous definition of separability and diversity of skills at the beginning, accompanied by a few sentences dedicated to motivation and examples", and this will be done in the revision. Since there would be a trade-off between theoretical and experimental results, we will carefully consider how to integrate some experiments without compromising the theoretical emphasis.
> > >
> > > ---
> > >
> > > Thanks again for your support and thoughtful advice.

---

> > > > ### Comment · Reviewer_ZXDZ · 2023-11-23
> > > >
> > > > Thank you for your detailed response!
> > > >
> > > > It helps me very much to understand the details in the paper further. I raised my score to 8.

---

### Official Review · Reviewer_xY4U · 2023-11-04

**Soundness:** 3 good
**Presentation:** 3 good
**Contribution:** 3 good
**Rating:** 8
**Confidence:** 4

**Summary:**

This paper analyzes unsupervised skill-learning through a rigorous mathematical lens, focusing on the properties of the learned skills and their usefulness for downstream tasks. Most existing works use mutual information as the skill-learning objective and Eysenbach et al. (2022) provides a mathematical analysis of the same. This work analyzes the mutual information-based skill learning paradigm with a focus on downstream task adaptability via worst-case adaptation cost. This work also introduces a complementary metric called LSEPIN to measure diversity of learned skills. The authors show that maximizing MI or LSEPIN is essentially optimizing the KL divergence between state distributions. As an alternative, they suggest using the Wasserstein metric owing to it being a proper metric, and propose a new skill learning objectives built upon Wasserstein distance.

**References:**

Benjamin Eysenbach, Ruslan Salakhutdinov, and Sergey Levine. The information geometry
of unsupervised reinforcement learning. In The Tenth International Conference on Learning
Representations, ICLR 2022, Virtual Event, April 25-29, 2022. [OpenReview.net](http://openreview.net/), 2022. URL
https://openreview.net/forum?id=3wU2UX0voE.

**Strengths:**

There is a long line of work on unsupervised skill learning based on mutual information maximization between states and skills, most of the work being motivated by intuition and empirical performance. This work complements Eysenbach et al. (2022) by providing a rigorous understanding of the properties of the learned skills and provides useful insights. The analysis presented in this work is novel, to the best of my knowledge and comprises a fairly significant advancement of our understanding of this sub-area of RL. The quality of analysis and writing is satisfactory, with sufficient background and context provided before explaining the main results of the paper.

**References:**

Benjamin Eysenbach, Ruslan Salakhutdinov, and Sergey Levine. The information geometry
of unsupervised reinforcement learning. In The Tenth International Conference on Learning
Representations, ICLR 2022, Virtual Event, April 25-29, 2022. [OpenReview.net](http://openreview.net/), 2022. URL
https://openreview.net/forum?id=3wU2UX0voE.

**Weaknesses:**

This is a fairly strong submission which checks all the boxes. The only minor complaint is that the empirical results in Appendix H should ideally be a part of the main paper, since including them makes the submission more well-rounded and gives empirical validation for the results presented in Section 3.

**Questions:**

None

---

> ### Author Response · Authors · 2023-11-15
> **A response**
>
> We sincerely appreciate your time and attention and thank you for your valuable feedback! We will consider your suggestion for the presentation.

---

### Meta-Review · Area_Chair_5hgg · 2023-12-14

**Metareview:**

This paper provides theoretical analysis for Unsupervised Reinforcement Learning (URL) and proposes new metric LSEPIN and new objective WSEP, which have better theoretical properties than commonly used Mutual Information Skill Learning (MISL). The theoretical analysis is validated on a few environments. This paper provides one of the earliest analysis to an important sub-field of RL. The quality of analysis and writing is satisfactory. We thus recommend acceptance.

**Justification For Why Not Higher Score:**

Empirical validation can be made more comprehensive.

**Justification For Why Not Lower Score:**

The paper addresses an important problem. Quality is great.

---

### Decision · Program_Chairs · 2024-01-16

Accept (spotlight)